# Extreme Risk Mitigation in Reinforcement Learning using Extreme Value Theory

**Karthik Somayaji NS**                                                  *karthi@ucsb.edu*
*Department of Electrical and Computer Engineering*
*University of California Santa Barbara*

**Yu Wang**                                                              *yu95@ucsb.edu*
*Department of Electrical and Computer Engineering*
*University of California Santa Barbara*

**Malachi Schram**                                                       *schram@jlab.org*
*Thomas Jefferson National Accelerator LaboratoryS*

**Jan Drgona**                                                           *jan.drgona@pnnl.gov*
*Pacific Northwest National Laboratory*

**Mahantesh Halappanavar**                                               *hala@pnnl.gov*
*Pacific Northwest National Laboratory*

**Frank Liu**                                                            *fliu@odu.edu*
*School of Data Science*
*Old Dominion University*

**Peng Li**                                                              *lip@ucsb.edu*
*Department of Electrical and Computer Engineering*
*University of California Santa Barbara*

**Reviewed on OpenReview:** *https://openreview.net/forum?id=098mb06uhA*

## Abstract

Risk-sensitive reinforcement learning (RL) has garnered significant attention in recent years due to the growing interest in deploying RL agents in real-world scenarios. A critical aspect of risk awareness involves modelling highly rare risk events (rewards) that could potentially lead to catastrophic outcomes. These infrequent occurrences present a formidable challenge for data-driven methods aiming to capture such risky events accurately. While risk-aware RL techniques do exist, they suffer from high variance estimation due to the inherent data scarcity. Our work proposes to enhance the resilience of RL agents when faced with very rare and risky events by focusing on refining the predictions of the extreme values predicted by the state-action value distribution. To achieve this, we formulate the extreme values of the state-action value function distribution as parameterized distributions, drawing inspiration from the principles of extreme value theory (EVT). We propose an extreme value theory based actor-critic approach, namely, Extreme Valued Actor-Critic (EVAC) which effectively addresses the issue of infrequent occurrence by leveraging EVT-based parameterization. Importantly, we theoretically demonstrate the advantages of employing these parameterized distributions in contrast to other risk-averse algorithms. Our evaluations show that the proposed method outperforms other risk averse RL algorithms on a diverse range of benchmark tasks, each encompassing distinct risk scenarios.

# 1 Introduction

In the recent years, there has been a wide array of research in leveraging reinforcement learning (RL) as a tool for enabling agents to learn desired behaviors with safety guarantees (Gattami et al., 2021; Eysenbach et al., 2017; Pinto et al., 2017; Smirnova et al., 2019; Xu & Mannor, 2010). Risk averse RL involves training an RL agent to optimize a risk measure unlike risk neutral RL where agents are trained to maximize the expected value of future discounted rewards. In risk averse RL, the accurate quantification of risk is particularly crucial in preventing catastrophic failures and finds relevance in safety-critical domains such as accelerator control (Rajput et al., 2022), finance (Daluiso et al., 2023), and robotics (Pan et al., 2019), where agents must navigate risky or hazardous states.

In risk averse RL, risk measures play a vital role in assessing the uncertainty and the potential negative consequences associated with an agent's decisions. Risk-averse RL algorithms employing various risk measures often consider the distribution of returns to quantify risk. Among these risk measures, Conditional Value at Risk (CVaR) is widely used, relying on the expected value of extreme quantiles of the return distribution to quantify risk (Tamar et al., 2014a; Hiraoka et al., 2019). In contrast, expectiles, which minimize the expected absolute deviation from the target, offer a robust estimate of central tendency (Marzban et al., 2021). Simpler methods for quantifying risk, such as estimating the variance of the return distribution (Xia, 2016; PrashanthL. & Fu, 2018) are also utilized.

These conventional approaches often require modelling the entire distribution of returns (state action value distribution) to accurately calculate the risk measure. In risk-averse RL, the focus is on estimating and quantifying low-probability risks, where the distribution of returns may display extreme skewness or heavy tails. Conventional methods model the distribution of returns typically through sampling, a data-intensive process that raises concerns about the accuracy of the modelled distribution, especially in low probability regions. Furthermore, in scenarios where collecting many samples is restrictive, the tail regions of the distribution of returns may not be accurately represented. Percentile risk measures, such as CVaR (Tamar et al., 2014a; Hiraoka et al., 2019), demand precisely modelled tail regions of the distribution of returns for accurate risk measure estimations. When the extreme tail regions of the distribution are inaccurately modeled, the risk measure becomes unreliable and may compromise the risk aversion of the RL agent. Thus, in risk averse RL, accurately modelling the tail region of the state action value distribution is key to good risk averse behavior. Extremely infrequent risk events, labeled as extreme risks (Troop et al., 2019), naturally exhibit low-probability skewed tails. Additionally, in situations where sample collection is expensive and adequate samples from the tail are lacking, traditional risk-averse RL algorithms face challenges such as high-variance risk measure estimates which are unreliable for optimization, as noted in (Pan et al., 2020; Beranger et al., 2021). These challenges may lead risk-averse RL algorithms to overlook extreme risk scenarios potentially resulting in catastrophic consequences in safety-critical applications.

We provide a simple demonstration on how the high variance in risk measure estimates, can affect risk aversion. We use the Half-Cheetah environment (Brockman et al., 2016) for our demonstration. Following the setup of (Urpí et al., 2021), we devise a scheme where the agent is penalized rarely for moving over a certain velocity threshold. We provide all details on implementation and set up in Section 8. The rareness of the penalty is controlled using a Bernoulli random variable which penalizes the reward with a Bernoulli penalization rate $\mathbf{p}$. We train risk averse RL agents based on (Urpí et al., 2021) and our proposed method, EVAC, using the CVaR risk measure (Tamar et al., 2014a) for different rare risk scenarios $\mathbf{p} = 5\%, 15\%$ and ascertain the percentage of times the Half-Cheetah agent crosses the threshold velocity in an episode. As $\mathbf{p}$ decreases the penalties become rarer and the risks become more extreme. We illustrate the percentage of crossing the threshold velocity as a function of the Bernoulli penalization rate in Figure 1a.

In Figure 1c, we present the empirical distribution and the modeled distribution according to (Urpí et al., 2021) for a specific state-action pair, depicted through their cumulative distribution functions (CDF). It is noted that under extreme rare risk conditions (at $\mathbf{p} = 5\%$), the tail of the modeled distribution (illustrated in red) does not accurately represent the true empirical distribution (illustrated in blue). Furthermore, Figure 1b includes a depiction of the variance associated with the risk measure, Conditional Value at Risk (CVaR), for the distribution modeled by (Urpí et al., 2021) and our proposed method, EVAC. We observe from Figure 1a and Figure 1b that higher variance of the risk measure estimate leads to poor risk aversion

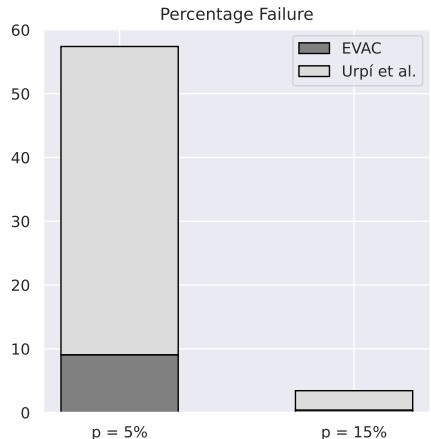

(a) Risk aversion as a function of extreme risk in (Urpí et al., 2021) and EVAC

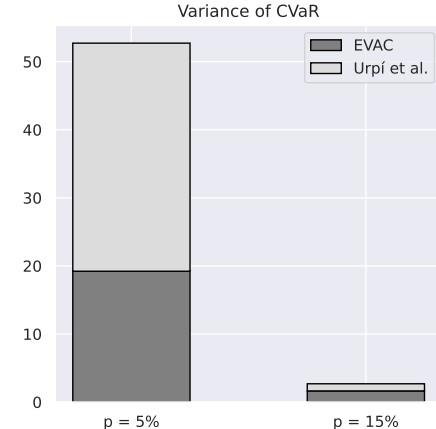

(b) Variance of the CVaR as a function of extreme risk in (Urpí et al., 2021) and EVAC

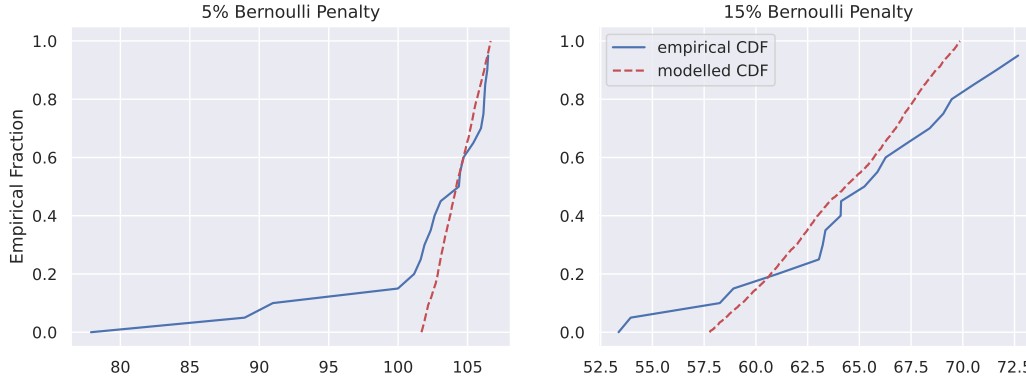

(c) Comparison between the empirical distribution of returns and the modelled distribution for extreme risk (5% Bernoulli penalty) and moderate risk (15% Bernoulli penalty) in (Urpí et al., 2021)

Figure 1: Risk aversion and distribution modelling performed by (Urpí et al., 2021) as a function of extreme risk: As the risks become extremely rare, traditional risk averse RL methods face challenges in modelling the tail distribution and also exhibit higher variance in the risk measure estimate. Thus, they may end up overlooking risky states and exhibit poor risk aversion.

generally. Under conditions of extreme rare risks, conventional risk-averse methodologies tend to display an increased variance in the estimate of the risk measure in comparison to EVAC. The high variance in the risk measure estimates may equally overestimate or underestimate the risk and obstruct the policy from effectively converging towards a risk-averse policy. As shown in Figure 6a, traditional risk-averse reinforcement learning (RL) methods tend to overlook extremely rare risky events (with a 5% Bernoulli penalization rate), leading to inadequate risk aversion. However, as the penalty frequency increases (to a 15% Bernoulli penalization rate), the variance of the risk measure drops and the risk-averse RL algorithm shows improved risk aversion capabilities. Consequently, the failure to accurately model the tail in scenarios of extreme and rare risk can result in suboptimal risk aversion.

Thus, there is an acute need to improve risk aversion by accurately modelling the prediction of extreme values of the return (state-action value) distribution. In this work, we propose to develop methods that help in extreme risk mitigation by modelling the state-action value distribution accurately. In particular, we develop EVAC, an extreme value theory (EVT) based method to model the quantiles of the tail region of the state

action value distribution. Extreme value theory (EVT) posits that the tail distribution follows a certain parametric form, which we propose to fit for the particular case of the state action value distribution. We theoretically and empirically demonstrate the reduction in the variance of the risk measure while employing EVAC which leads to better risk aversion even under extreme rare risks. Thus, our contributions are that:

1. We recognize the challenges faced by conventional distribution-based RL methods (Ma et al., 2020; Dabney et al., 2017; Urpí et al., 2021) in effectively modelling and mitigating extremely rare risks.

2. We propose a novel approach to model the tail region of the state-action value distribution, inspired by extreme value theory (EVT) by utilizing a General Pareto distribution. Importantly, we demonstrate a reduction in variance in the estimation of the quantiles of the tail distribution.

3. We propose a novel actor-critic distributional RL algorithm called Extreme Valued Actor-Critic (EVAC) and conduct comprehensive empirical evaluations of our proposed method across various extreme risk environments. We compare its risk aversion and performance against widely used baselines.

## 2 Related Work

There has been an extensive study in incorporating reinforcement learning in risk avoidance. (Chow & Ghavamzadeh, 2014; Tamar et al., 2014b; Chow et al., 2015b) have studied quantification of risk in terms of percentile criteria, specifically the CVaR (conditional value at risk). Other risk averse measures also include (Xia, 2016; PrashanthL. & Fu, 2018) which incorporate variance as a risk measure, while (Chow et al., 2015a) uses a notion of range to discuss risk. All the above metrics do require an understanding of the distribution of the quantity of interest. Most works in the RL context deal with risk as it is connected to the distribution of the state action value distribution (or the Q-function). The work of (Bellemare et al., 2017) offers a distributional perspective on the value functions. (Dabney et al., 2017) approximates the distribution of the value functions in terms of the quantiles of the distribution. Other works like (Ma et al., 2020; Urpí et al., 2021; Tang et al., 2019) use the notion of quantiles of the value function distribution to quantify risk and (Srinivasan et al., 2020) also discusses methods for fine tuning agents in transfer learning scenarios.

(Liu et al., 2022) perform post-hoc risk analysis of players in a sports game while accounting for epistemic uncertainty. They use a distributional RL framework as a policy evaluation tool to evaluate player's performance. Specifically, they employ an offline dataset for performance prediction of a player and assess risky behavior using quantile based distributional modeling. Furthermore to account for distribution shift between the offline dataset and the test dataset, the work employs epistemic uncertainty estimation via density estimation for calibration. (Ma et al., 2021) is a distributional RL framework used for risk aversion in the offline learning paradigm. Compared to the offline risk averse setting of (Urpí et al., 2021) namely (O-RAAC), (Ma et al., 2021) penalize the quantiles of OOD state action pairs pointwise. They demonstrate superior risk aversion in the offline setting in comparison to other offline risk averse RL methods. (Greenberg et al., 2022) propose a method to balance high return policies against risk averse policies. They introduce a soft risk mechanism and a cross-entropy method to enhance sample efficiency and maintain risk aversion. (Hau et al., 2022) consider alternate risk measures such as the entropic risk measure and the entropic value at risk in tabular settings for robust risk averse applications, where the agent needs to mitigate performance loss in uncertain states. Similar in spirit to (Liu et al., 2022), they combine epistemic uncertainty estimates with these risk measures along with a dynamically changing risk level to achieve risk aversion and robustness to model uncertainty. (Killian et al., 2023) operate in the offline RL setting and investigate estimating the risk associated with a policy in terms of the expected worst case outcomes. The goal is to develop a distributional value function to provide an early warning of risk over possible future outcomes. They use separate dead end and terminal state reward signals to construct separate MDPs to derive two value distributions. A state action pair is classified a dead end if the CVaR of both the value distributions lies below a given threshold. (Luo et al., 2023) study risk aversion as it corresponds to avoidance of highly noisy states. They lay out foundations on why the variance as a risk measure falls short for risk averse policy estimation. They propose Gini deviation, a new risk measure that overcomes the shortcomings of the variance risk measure and also demonstrates good risk aversion.

While numerous studies have explored risk aversion using various risk measures and techniques, there has been minimal focus on increasing the accuracy of the modeled return distributions that these risk measures rely on, particularly in scenarios involving extreme and rare risks.

When addressing rare risky events, extreme value theory (EVT) (Haan & Ferreira, 2006) provides a framework to characterizing the asymptotic behavior of the distribution of the sample extrema. EVT finds extensive application in modelling rare events across domains from finance, to operations research and meteorology to multi-armed bandit problems (Roncalli, 2020; Santiñaque et al., 2023; Can et al., 2023; Troop et al., 2019). In the reinforcement learning context, (Garg et al., 2023) recently discusses the use of EVT in estimating the maximum of the Q-value in the Bellman update. Our work is different from (Garg et al., 2023) on three main fronts. **(i)** Firstly, (Garg et al., 2023) operates in the non-distributional RL setting and aims to learn the optimal value function in the max-entropy RL, inspired by principles in EVT. In max-entropy RL, the optimal Bellman operator requires estimating the optimal value function, which is intractable in continuous action spaces. (Garg et al., 2023) uses EVT inspired Gumbell regression to compute the optimal value function in the non-distributional max-entropy RL setting. However, our work operates in the distributional RL setting and aims to perform risk averse decision making in extremely rare risky scenarios by modelling the extreme quantiles of the state action value distribution. Particularly, our work draws inspiration from EVT to model the entire distribution of the state action value distribution unlike (Garg et al., 2023). **(ii)** (Garg et al., 2023) uses the fact that the sample maxima can be characterized by the Gumbell distribution to estimate the max entropy optimal value function. However, our work considers modelling the entire state action value distribution precisely by using the principle of asymptotic conditional excess distributions to estimate the underlying tail behavior of the state action value distribution. **(iii)** (Garg et al., 2023) uses Gumbell regression as a tool to define and train a soft Q-function by gathering inspiration from the Gumbell distributed Bellman errors. This is accomplished by using the Fisher Tippet Theorem (Theorem 4.1) which provides the limiting behavior of sample maxima. In our work, we estimate the realizations of the critic (state action value) distribution over a threshold by employing another key theorem in extreme value theory namely the Pickands-Balkema-de Haan Theorem (Theorem 4.2). Particularly, our method uses maximum likelihood estimation of the GPD distribution to estimate the state action value distribution.

## 3 Notations

In this rest of the paper, we adopt the notation for the standard Markov Decision Process (MDP) characterized by the tuple $(\mathcal{S}, \mathcal{A}, P_R, P_S, \gamma)$, where $\mathcal{S}$ is the state space, $\mathcal{A}$ is the action space, $P_R$ is the stochastic reward kernel such that $P_R : \mathcal{S} \times \mathcal{A} \to \mathcal{P}(\mathcal{R})$, where $\mathcal{R}$ is the reward set. $P_S : \mathcal{S} \times \mathcal{A} \to \mathcal{P}(\mathcal{S})$ is the probabilistic next state transition kernel, and $\gamma$ is the discount factor. The policy $\pi$ of the agent is a mapping $\pi : \mathcal{S} \to \mathcal{P}(\mathcal{A})$. We denote $S_t, A_t, R_t$ as the encountered state, action and reward, respectively at time step $t$. The future sum of discounted returns is a random variable denoted by $J^\pi(s, a) = \sum_{t=0}^{\infty} \gamma^t R_t$, where $R_t \sim P_R(S_t, A_t)$ and $A_t \sim \pi(S_t)$ with $S_0 = s; A_0 = a$. We denote the distribution corresponding to the random variable $J^\pi$ as $Z^\pi$. In this work, we primarily operate on stationary environments with dense rewards and continuous action space settings.

## 4 Background

### 4.1 Distributional Reinforcement Learning

Distributional reinforcement learning (distributional RL) Bellemare et al. (2017); Dabney et al. (2017) entails the modelling of the complete distribution of the state-action value function. In contrast to traditional RL which focuses solely on modelling the expected value of the state-action value function's distribution as a point estimate, distributional RL aims to model the entire distribution of the state action value function.

The state action value distribution $Z$ under policy $\pi$ is updated using the distributional Bellman operator

$$T^\pi Z^\pi(s, a) = r(s, a) + \gamma \mathrm{E}_{s' \sim P_S(s,a), a' \sim \pi(s')} Z^\pi(s', a') \tag{1}$$

The Bellman operator $T^\pi : \mathcal{P}(\mathrm{R}^{\mathcal{S} \times \mathcal{A}}) \to \mathcal{P}(\mathrm{R}^{\mathcal{S} \times \mathcal{A}})$ operates on the space of probabilities over the reals R, for each state action pair. Focusing attention on Eqn.1 reveals that the update of the LHS distribution $Z^\pi(s, a)$ happens via sampling the mixture distribution $\mathrm{E}_{s' \sim P_S(s,a), a' \sim \pi(s')} Z^\pi(s', a')$, scaling it by $\gamma$ and shifting it by $r(s, a)$, which is the sampled scalar reward. Thus, the update of $Z^\pi(s, a)$ can be viewed as a scale and shift operation of the mixture distribution $\mathrm{E}_{s' \sim P_S(s,a), a' \sim \pi(s')} Z^\pi(s', a')$. The distribution function $Z^\pi(s, a)$ characterizes the values that the random variable $J^\pi(s, a)$ can assume. Thus, knowledge of the distribution function $Z^\pi(s, a)$ aids in understanding the extreme values that $J^\pi(s, a)$ can be assigned. In risk averse RL, state action pairs whose distributions $Z^\pi(s, a)$ assume low extremal values denote risky states and need to be avoided. Thereby, distributional RL provides a tool to quantify the uncertainty and risk for risk averse applications.

## 4.2 Quantile Regression

One of the popular methods to model the distribution of the state-action value function in distributional RL is through the quantiles of the distribution. The quantiles of the distribution are often estimated through the quantile regression framework (used in risk averse RL applications including Ma et al. (2020); Dabney et al. (2017); Urpí et al. (2021)). Quantile regression proposed by Koenker & Bassett Jr (1978) estimates the true quantile value of a distribution by minimizing the pinball loss. Assume a random variable $Y$ with its unknown true distribution function $F_Y(.)$ and probability density function $f_Y(.)$. The goal lies in estimating the true quantile values of $F_Y$ denoted by $\theta_\tau$ for $\tau \in [0, 1]$, the quantile level. The quantile predicted by the quantile regression framework is a unique minimizer of the pinball loss $\mathcal{L}(\theta_\tau)$ given by

$$\mathcal{L}(\theta_\tau) = \mathrm{E}_{y \sim F_Y}(y - \theta_\tau)(\tau - 1_{y - \theta_\tau < 0}). \tag{2}$$

In deep reinforcement learning, a modified smooth loss function called the empirical quantile Huber loss Huber (1964) is instead used for better gradient back propagation.

## 4.3 Extreme Value Theory (EVT)

Modeling the extreme values of distributions under low data availability is challenging. Extreme Value Theory (EVT) is a statistical framework that focuses on modelling and analyzing the tail behavior of probability distributions, particularly the distribution of extreme values. It provides methods to characterize and predict rare and extreme events, making it valuable in assessing and managing risks associated with tail events. We formally introduce the two main theorems in EVT below.

**Theorem 4.1 (Fisher Tippet Theorem Basrak (2011))** *Let $X_1, \cdots, X_n$ be a sequence of IID random variables, with a distribution function (CDF) $F$. Let $M_n$ represent the sample maximum. If there exist constants $a_n > 0$ and $b_n$ and a non-degenerate distribution function $G$ such that:*

$$\lim_{n \to \infty} P\left\{ \frac{M_n - b_n}{a_n} \leq x \right\} = G(x),$$

*then the distribution function $G(x)$ is called the Generalized Extreme Value distribution (GEV) and can be decomposed into either the Gumbell, Frechet or the Weibull distribution.*

Intuitively the Fisher Tippet Theorem describes that the normalized sample maxima of a distribution $F$, converges in distribution to the GEV distribution.

The other central theorem in EVT is the Pickands-Balkema -de Haan Theorem (PBD Theorem) which inspects the conditional exceedance probability above a threshold $u$, of a random variable $X$ with a distribution function $F$..

**Theorem 4.2 (Pickands-Balkema-de Haan Theorem )** *Pickands III (1975) Let $X_1 \cdots X_n$ be a sequence of IID random variables with distribution function (CDF) given by $F$ whose limiting behavior approaches the GEV distribution. Let $F_u(x) = P(X - u \leq x | X > u)$ be the conditional excess distribution. Then,*

$$\lim_{u \to \infty} F_u(x) \xrightarrow{D} H_{\xi, \sigma}(x),$$

where $H_{\xi,\sigma}(x)$ is the Generalized Pareto distribution (GPD) with parameters $\xi, \sigma$.

Intuitively Theorem4.2 describes that the conditional distribution $F_u$ of the random variable $X$ approaches the GPD distribution for large enough $u$. The CDF of the GPD distribution $F_{\xi,\sigma}(x)$ is given by:

$$\begin{cases} 1 - \left(1 + \frac{\xi x}{\sigma}\right)^{-1/\xi} \text{ for } \xi \neq 0 \\ 1 - \exp(-\frac{x}{\sigma}) \text{ for } \xi = 0 \end{cases} \tag{3}$$

## 5 Motivation

Modelling the quantiles of the state-action value distribution, especially the low-probability extreme realizations (tail quantiles) is crucial in risk averse RL. However, this is challenging when the extreme tail regions of the underlying distribution are skewed or heavy tailed. This challenge arises as the traditional sampling based estimations of these quantiles tend to exhibit high variance, leading to potential compromises in risk aversion strategies. To address this issue, our approach in this study involves leveraging extreme value theory-based parameterized asymptotic distributions to effectively model the low-probability tail regions of the state-action value distribution.

To underscore the importance of employing extreme value theory-based parameterized distributions for tail distribution characterization, we initially examine the challenges associated with utilizing sampling based methods to estimate the quantiles of the tail distribution through quantile regression.

### 5.1 Challenges when employing the sampling distribution in quantile regression

Assume a random variable $Y$ with its unknown true distribution function $F_Y(.)$ and probability density function $f_Y(.)$ Assume $N$ samples sampled from $F_Y$, $\{y_1, y_2, \cdots, y_N\}$. The aim is to find the quantile value estimate $\theta_\tau^N$ using quantile regression, by minimizing the empirical pinball loss for a given quantile level $\tau$, which follows from Eqn.2:

$$\mathcal{L}(\theta_N^\tau) = \frac{1}{N} \sum_{i=1}^{N} (y_i - \theta_\tau^N)(\tau - 1_{y_i - \theta_\tau^N < 0}) \tag{4}$$

Importantly, the asymptotic convergence of the quantile regression estimator to the true quantile value is discussed in Koenker & Bassett Jr (1978) as

$$\sqrt{N}(\theta_\tau^N - \theta_\tau) \xrightarrow{D} \mathcal{N}(0, \tilde{\lambda}^2),$$

where the variance of the estimator $\theta_\tau^N$ is given by

$$\lambda^2 = \frac{\tilde{\lambda}^2}{N} = \frac{\tau(1-\tau)}{N \cdot f_Y^2(\theta_\tau)}. \tag{5}$$

The variance of the quantile regression estimate $\theta_\tau^N$ is dependent on the number of samples $N$ from the distribution that are used to estimate the quantiles and also on the squared probability density $f_Y^2(\theta_\tau)$ at the quantile value $\theta_\tau$.

Under a fixed number of samples, the variance of the quantile regression estimate increases in the case of rare extreme valued state-action value distributions when the density function $f_Y(\theta_\tau)$ assumes low probability values in the tail regions ($\tau \to 1^-$ i.e. heavy tails). Beranger et al. (2021); Pan et al. (2020) also discuss the estimation inaccuracy of the quantile estimates under lower data availability. Bai et al. (2021) specifically discusses the inherent undercoverage associated with quantile regression estimator for tail quantile regions. Such evidence coupled with the high variance property acts as a deterrent to exclusively choosing the sampling distribution for estimating extreme quantiles in the case of distributions with low probability extreme events.

Thereby, we investigate the modelling of the state action value distribution when the sampling distribution assumes rare extreme values.

### 5.2 EVT based modelling of the state-action value distribution

Given the naturally occurring heavy tails in rare extreme events, direct sampling from the return distribution (state-action value distribution), may yield insufficient information regarding extreme values. Consequently, we propose an approach that involves fitting a parameterized distribution to the tail regions of the state-action value distribution. This approach draws inspiration from the principles of Extreme Value Theory (EVT), which advocates the fitting of asymptotic distributions to the extreme tail regions of the state action value distribution.

## 6 Method

Extremely rare reward penalties like the ones discussed in Section 1 cause the state action value distribution to be characterized by low probability tails. The higher probability occurrences of the true random return $J^\pi(s, a)$ can be learnt reasonably well, unlike the low probability extreme valued realizations. Thereby, we propose to decompose the state action value distribution into two regions namely, the non-tail region and the tail region.

**Remark 6.0.1** *Although higher rewards are considered better in reinforcement learning, we negate the rewards to maintain consistent notation with literature in EVT. This does not affect any analysis. Because of the negated reward, we swap focus from the left tail of the state action value distribution to the right tail.*

In order to capture the extreme regions within the state-action value distribution, we depict the tail distribution and the non-tail distribution, in Figure 2. For a given threshold $u$, we denote by $Z_L^\pi(s, a)$, the non-tail distribution. The subscript '$L$' is used for denoting support values of $Z^\pi$ lower than $u$. Similarly, we denote by $Z_H^\pi(s, a)$, the tail distribution. The subscript '$H$' denotes distribution with support values higher than $u$, which is assumed to be a sufficiently high threshold for the state action pair $(s, a)$. We assume that the area under $Z_L^\pi(s, a)$ and $Z_H^\pi(s, a)$ to be 1. So, the state action value distribution $Z^\pi(s, a)$ is obtained by rescaling $Z_L^\pi(s, a)$ and $Z_H^\pi(s, a)$.

$$Z^\pi(s, a) = \eta \cdot Z_L^\pi(s, a) + (1 - \eta) \cdot Z_H^\pi(s, a) \tag{6}$$

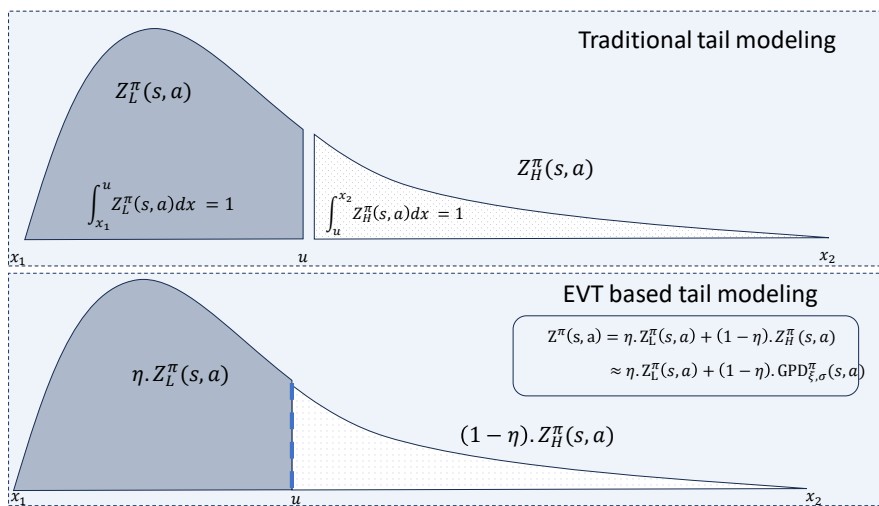

Figure 2: Modelling the tail and non-tail distributions of the state action value function. The area under the non-tail distribution $Z_L^\pi(s, a)$ and tail distribution $Z_H^\pi(s, a)$ is assumed to be 1.

The fundamental idea guiding our methodology for modelling extreme values within the state-action value distribution is to conceptualize the tail region of the distribution as the conditional excess distribution beyond a certain threshold. To formalize this, we invoke the Pickands-Balkema-de Haan theorem (Theorem 4.2) to approximate the tail regions of the state-action value distribution $Z^\pi$. According to Theorem 4.2, the

conditional excess can be expressed through the Generalized Pareto Distribution (GPD) with parameters $\xi, \sigma$. Consequently, for each state-action pair $(s, a) \in \mathcal{S} \times \mathcal{A}$ within the state-action value distribution $Z^\pi(s, a)$, the tail region can be effectively modeled by an associated GPD distribution with parameters $\xi(s, a), \sigma(s, a)$. Therefore, the tail regions of each $Z^\pi(s, a)$ can be viewed as parameterized GPD distributions in the $\mathcal{S} \times \mathcal{A}$ space, denoted as $\mathrm{GPD}\Big(\xi(s, a), \sigma(s, a)\Big)$ with distinct thresholds $u$ for each $(s, a)$. The threshold $u$ for each $(s, a)$ pair is the value of the support distribution $Z^\pi(s, a)$, where its CDF has a value of $\eta$. In the limit, the tail distribution $Z_H^\pi(s, a)$ can be represented by $\mathrm{GPD}\Big(\xi(s, a), \sigma(s, a)\Big)$ and appropriately rescaled to obtain $Z^\pi(s, a)$ as outlined in Eqn. 6.

Fitting the asymptotic GPD distribution enables effective extrapolation and the generation of new data points during sampling, which makes the EVT based tail modelling data efficient. The merit of this approximation lies in its ability to capture the extreme value behavior of $Z^\pi$ even with limited data availability, owing to the fitting of asymptotic Extreme Value Theory (EVT) distributions. We provide a detailed explanation of the GPD fitting procedure in Section 7 and the algorithmic flow of our approach in Algorithm 1.

### 6.1 Variance reduction in the quantile regression estimate using EVT based tail modelling

Having characterized the tail region of the state-action value distribution using the parameterized Generalized Pareto Distribution (GPD), our objective is to explore the implications of replacing the sampling distribution with the proposed parameterized state-action value distribution when performing quantile regression.

The proposed methodology involves updating quantiles of the state-action value distribution through quantile regression. As detailed in Section 5.1, the variance of the quantile regression estimator is influenced by the square of the probability density at a specific quantile level. This prompts a crucial inquiry: how does the variance of the quantile regression estimate change when utilizing a GPD approximation for the tail region of the underlying distribution? To address this question, we conduct an analysis of the variance of the quantile regression estimator for a distribution with a parameterized GPD tail and compare it to a distribution without such tail modeling.

Assume a random variable $Y$ with its distribution function $F_Y$. Let the $\tau^{th}$ quantile of $F_Y$ be denoted as $\theta_\tau$. For any sufficiently large quantile level $\eta \in [0, 1)$, and a smaller increment quantile level $t \in [0, 1 - \eta)$ such that, $\eta + t \in [0, 1)$, we have the corresponding quantiles of the distribution $\theta_\eta$ and $\theta_{\eta+t}$. We define the excess value of the quantile as $x = \theta_{\eta+t} - \theta_\eta$. We define the CDF of the GPD distribution by $F_{H_{\xi,\sigma}}$ and its density function by $f_{H_{\xi,\sigma}}$.

$$F_Y(\theta_{\eta+t}) = P\Big(Y \le \theta_\eta + x\Big) \tag{7}$$
$$= \eta + (1 - \eta)P\Big(Y - \theta_\eta \le x | Y > \theta_\eta\Big)$$
$$\approx \eta + (1 - \eta)F_{H_{\xi,\sigma}}(x)$$

As mentioned earlier for sufficiently large $\eta$, $P\Big(Y - \theta_\eta \le x | Y > \theta_\eta\Big)$ approaches the GPD distribution $F_{H_{\xi,\sigma}(x)}$ in the limit. Thus, we have

$$P(Y \le \theta_\eta + x) \approx \eta + (1 - \eta)P(X \le x) \tag{8}$$

where $X \sim H_{\xi,\sigma}$. It follows from Eqn.8, that,

$$f_Y(\theta_\eta + x) \approx (1 - \eta)f_{H_{\xi,\sigma}}(x) \tag{9}$$

If we represent the quantiles of $F_{H_{\xi,\sigma}}$ by $\theta^H$, then we have the following relationship between the quantiles of $F_Y$ and the quantiles of the GPD distribution :

$$\theta_{\eta+t} \approx \theta\eta + \theta^H_{\frac{t}{1-\eta}} \tag{10}$$

We are interested in representing the quantiles for sufficiently large $\eta$ and higher. Following Eqn. 5, the variance of the quantile regression estimator in estimating the quantile $\theta_{\eta+t}$ of the distribution function $Y$ is

$$\lambda_Y^2 = \frac{(\eta+t)(1-\eta-t)}{N \cdot (1-\eta)^2 f_{H_{\xi,\sigma}}^2\left(\theta_{\frac{t}{1-\eta}}^H\right)}$$

In Eqn. 10, $\theta_\eta$ corresponds to the $\eta^{th}$ quantile of the distribution corresponding to the threshold quantile level $\eta$. Assuming that the distribution has sufficient probability mass at quantile level $\eta$, $\theta_\eta$ may be accurately estimated with the provided samples. If $\theta_\eta$ is known, one may simply estimate the $\frac{t}{1-\eta}^{th}$ quantile of the GPD distribution and shift it by $\theta_\eta$. Thus the quantile regression estimator's variance in estimating $\eta + t$ quantile of $Y$ using the GPD is given by:

$$\lambda_H^2 = \frac{(t/(1-\eta))(1-t/(1-\eta))}{N \cdot f_{H_{\xi,\sigma}}^2\left(\theta_{\frac{t}{1-\eta}}^H\right)}$$

We can verify that $\lambda_Y^2 \gg \lambda_H^2$ for large values of $\eta$, e.g., close to 1.0. Therefore, we show that GPD based modelling of the tail region of the distribution $Y$ helps reduce the variance in the estimation of higher quantiles. We also illustrate this in Figure.3 with a few candidate values for $\eta = 0.75, 0.8, 0.85$.

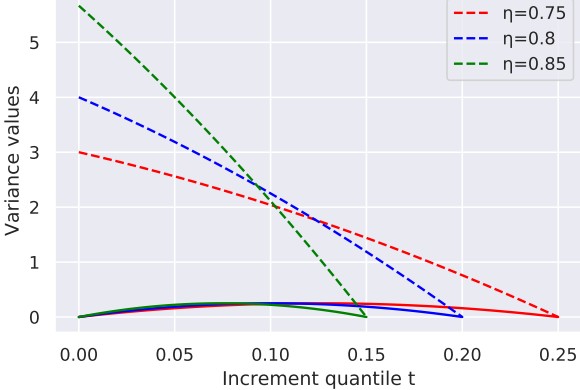

Figure 3: Dashed lines indicate $\lambda_Y^2$ and the solid lines indicate $\lambda_H^2$. $\lambda_H^2$ is always upper bounded by the $\lambda_Y^2$ which illustrates variance reduction while using GPD for approximating the tail.

The preceding analysis establishes that when $\eta$ is large enough (and the tail distribution converges to the GPD distribution), the variance of the quantile regression estimator with the parameterized state-action value distribution is smaller in comparison with the variance of an estimator with a non-parameterized state-action value distribution.

The reduced variability in the quantile regression estimate is beneficial as it also lowers the variability in the estimate of the risk measure. For instance, the Conditional Value at Risk (CVaR) risk measure, employed in optimizing risk-averse policies (as outlined in Equation 12), shows reduced variability when there is less variability in the tail quantile estimates. This decrease in risk measure variability can, in turn, expedite the process of developing risk-averse policies with a limited number of interactions with the environment.

# 7 Extreme Valued Actor Critic (EVAC)

## 7.1 Actor Critic Framework

To scale the GPD modelling to large and continuous state action spaces, we introduce an actor-critic deep reinforcement learning method, named Extreme Valued Actor-Critic (EVAC). This approach utilizes parameterized Extreme Value Theory (EVT) based distributions to represent the state-action value distribution. Aligned with existing frameworks like Lillicrap et al. (2016); Fujimoto et al. (2018); Urpí et al. (2021), our architecture comprises an actor and critic network. Similar to Urpí et al. (2021), the actor network is employed for optimizing the risk measure and suggesting optimal risk-averse actions. Conversely, the critic network is dedicated to modelling the quantiles of the state-action value distribution. Specifically, the critic network is utilized to capture both the quantiles of the non-tail region and those of the EVT-based parameterized tail distribution. We present the framework for training the critic that models the state-action value distribution.

## 7.2 Training the state-action value distribution

The state-action value distribution models the distribution of the future returns and is updated using the Bellman update :

$$T^\pi Z^\pi(s,a) = r(s,a) + \gamma\Big[\eta \cdot Z_L^\pi(s',a') + (1-\eta) \cdot Z_H^\pi(s',a')\Big] \tag{11}$$

where $(s,a,r,s',a')$ represent the state, action, reward, next-state and next-action tuple. The distribution $Z^\pi(s',a') = \eta \cdot Z_L^\pi(s',a') + (1-\eta) \cdot Z_H^\pi(s',a')$ is decomposed into the non-tail distribution $Z_L^\pi(s',a')$ and the tail distribution $Z^\pi(s',a')$ as in Eqn 6. In EVAC, we propose learning the state action value distribution $Z^\pi(s,a)$ and the parameters of the GPD distribution $\text{GPD}(\xi(s,a),\sigma(s,a))$.

The state action value distribution $Z^\pi(s,a)$ is modelled using quantiles. We denote the $\tau^{th}$ quantile of $Z^\pi(s,a)$ as $Z^\pi(s,a)|_\tau$. Firstly, the non-tail distribution function $Z_L^\pi(s,a)$ is defined as:

$$Z_L^\pi(s,a)|_{\frac{\tau}{\eta}} = Z^\pi(s,a)|_\tau, \forall \tau \in [0,\eta]$$

The GPD distribution with parameters $\xi(s,a), \sigma(s,a)$ is denoted as $\text{GPD}(\xi(s,a),\sigma(s,a))$. The tail distribution $Z_H^\pi(s,a)$ is obtained by a shifted version of the GPD distribution and may be represented as

$$Z_H^\pi(s,a) = Z^\pi(s,a)|_\eta + \text{GPD}\Big(\xi(s,a),\sigma(s,a)\Big)$$

where $Z^\pi(s,a)|_\eta$ is the scalar shift and represents the threshold quantile of the state action value distribution $Z^\pi(s,a)$ corresponding to quantile level $\eta$.

Our aim is to model the quantiles of $Z^\pi(s,a)$ accurately by utilizing samples from the non-tail distribution function $Z_L^\pi(s',a')$ and the tail distribution $Z_H^\pi(s',a')$. Our proposed training procedure to train the state action value distribution $Z^\pi(s,a)$ in Eqn 11 encompasses the following distinct components:

- Sampling from the non-tail distribution $(Z_L^\pi(s',a'))$ and the tail distribution $(Z_H^\pi(s',a'))$.

- Quantile regression to estimate the quantiles of the state action value distribution $Z^\pi(s,a)$.

- Updating the parameters $(\xi(s,a),\sigma(s,a))$ of the GPD distribution which controls the tail distribution $Z_H^\pi(s,a)$.

### 7.2.1 Sampling the non-tail distribution function and the tail distribution

Our goal is to estimate the quantiles of the state action value distribution $Z^\pi(s,a)$ of the current state action pair $(s,a)$ in Eqn 11 through quantile regression. In order to do so, we need to obtain samples from the RHS of Eqn 11 by sampling the tail distribution $Z_H^\pi(s',a')$ with proportion $(1-\eta)$ and the non-tail distribution $Z_L^\pi(s',a')$ with proportion $\eta$.

We obtain samples from the non-tail distribution $Z_L^\pi(s', a')$ using inverse transform sampling. For sampled quantile levels $\tau \sim U(0, \eta)$, the queried samples $Z^\pi(s', a')|_\tau$ correspond to sampling from the non-tail distribution $Z_L^\pi(s', a')$.

We obtain samples from the tail distribution $Z_H^\pi(s', a')$ by querying the GPD model. Specifically $Z_H^\pi(s', a') = Z^\pi(s', a')|_\eta + \text{GPD}\Big(\xi(s', a'), \sigma(s', a')\Big)$. First, we sample from the GPD distribution $\text{GPD}\Big(\xi(s', a'), \sigma(s', a')\Big)$ and shift it by the scalar threshold quantile value $Z^\pi(s', a')|_\eta$.

The shifted GPD distribution models the tail region of the state action value distribution, while the samples from $Z_L^\pi(s', a')$ model the non-tail region. Having obtained samples from the RHS of Eqn.11, the state action value function $Z^\pi(s, a)$ can be trained using quantile regression. It is to be noted that although the state action value function is modelled by quantile regression, our sampling procedure involves sampling of $Z_L^\pi(s', a')$ and the shifted GPD distribution which more accurately models the tail behavior, unlike Ma et al. (2020); Dabney et al. (2017); Urpí et al. (2021).

### 7.2.2 Quantile regression to update quantiles of $Z^\pi(s, a)$

The samples obtained from the tail distribution, $Z_H^\pi(s', a')$ (with proportion $1 - \eta$) and the non-tail distribution $Z_L^\pi(s', a')$, (with proportion $\eta$), are used to estimate the quantiles of the state action value distribution $Z^\pi(s, a)$. Consider $N$ such samples from the tail and non-tail distribution, each represented as $y_i$. Particularly, to estimate the $\tau^{th}$ quantile $\theta_\tau$ of $Z^\pi(s, a)$, the pinball loss $\mathcal{L}(\theta^\tau)$ is minimized.

$$\mathcal{L}(\theta^\tau) = \frac{1}{N} \sum_{i=1}^{N} (y_i - \theta_\tau)(\tau - 1_{y_i - \theta_\tau < 0})$$

This procedure accurately models the quantiles of the state action value distribution $Z^\pi(s, a)$ by obtaining samples from both the non-tail and the tail distributions. Since the pinball loss is non-smooth, we employ a smooth approximation of the pinball loss namely, the Huber loss Huber (1964).

### 7.2.3 Updating the parameters of the GPD Distribution

The previous procedure uses samples from the tail and non-tail distribution to update quantiles of the state action value distribution. However, the tail distribution $Z_H^\pi(s, a)$ which is modelled by the shifted GPD distribution itself needs to be updated. Thereby, it is imperative to appropriately fit the parameters $\xi(s, a), \sigma(s, a)$ of the Generalized Pareto Distribution (GPD) to accurately model Eqn. 11. The GPD is used to model the distribution of excess values above a given threshold (threshold quantile value).

$$\text{GPD}\Big(\xi(s, a), \sigma(s, a)\Big) = Z_H^\pi(s, a) - Z^\pi(s, a)|_\eta$$

where $Z^\pi(s, a)|_\eta$ is the threshold quantile. To acquire these excess values, we initially generate samples $Z^\pi(s, a)|_{\tau=\eta}^1$ and subtract them from the threshold $Z^\pi(s, a)|_{\tau=\eta}$ to obtain the surplus beyond the threshold. Subsequently, the GPD parameters $\xi(s, a), \sigma(s, a)$ can be determined through maximum likelihood estimation of the GPD distribution applied to the samples of excess over the threshold.

Thus, we provide a novel framework for using extreme value theory for state action value distribution estimation. In Section A, we provide a proof of the convergence of the Bellman operator in Eqn 11. In Section B, we provide details of the MLE estimation procedure to update parameters of the GPD distribution.

### 7.3 Policy Optimization

Once the critic, (encompassing both the tail and non-tail segments) has been trained under a fixed policy $\pi$ to produce the state action value distribution $Z^\pi(s, a)$, our focus shifts towards the actor's policy optimization, aimed at achieving a risk-averse behavior. In order to train risk averse policies, extreme values of the state-action value distribution are employed to guarantee optimal worst case performance. We propose to

employ the CVaR risk measure Tamar et al. (2014b); Ying et al. (2022) on the state-action value distribution $Z^\pi(s, a)$ for mitigating extreme risk. The CVaR (Conditional value at risk), is a risk measure that denotes the average worst case performance by integrating the quantiles of the state action value distribution between quantile levels $x_1$ and 1.0. The optimal policy $\pi^*$ in Eqn.12 is obtained through policy gradients over the CVaR. We choose $x_1 = 0.95$ in all experiments. ( Note the negation of the reward (Remark 6.0.1) which paves the way for higher values of $x_1$).

We expect that for a given amount of collected data, lower variance estimates of the CVaR for a given $(s, a)$ pair lead to better risk averse policies. We evaluate the both the variance and consequently the quality of the risk averse policies produced by EVAC and other baselines in the next section.

$$\pi^* = \arg\min_{\pi} \text{CVaR}(x_1) \tag{12}$$
$$= \arg\min_{\pi} \frac{1}{x_2 - x_1} \int_{\tau = x_1}^{1} Z\Big(s, \pi(s)\Big)\Big|_\tau d\tau$$

### 7.4 Algorithm for EVAC

---

**Algorithm 1:** Extreme Valued Actor Critic: EVAC

---

**Input:** Initialize GPD parameters $\{\sigma(s, a), \xi(s, a)\}$ the critic $Z(s, a)$, policy $\pi(s)$ and threshold quantile level $\eta$
**Policy Iteration:**
1. Sampling from the parameterized distribution and updating the critic (Section 7.2.1 and 7.2.2)
**for** *tuple* $(s, a, r, s', a' = \pi(s'))$ **do**

    $x \sim GPD\big(\xi(s', a'), \sigma(s', a')\big)$
    Define $Z'_H = Z(s', a')|_{\tau=\eta} + x$
    Define $Z'_L = Z(s', a')|_{\tau=\tau_0}$; where $\tau_0 \sim \text{Unif}(0, \eta)$
    Sample $p \sim \text{Bernoulli}_\eta$
    Define Bellman target: $Z_T = r + \gamma[\mathbf{1}_{p=0} Z'_H + \mathbf{1}_{p=1} Z'_L]$
    Update the critic $Z$ using samples $Z_T$ through quantile regression

2. Updating the GPD parameters $\xi(s, a), \sigma(s, a)$ (Section 7.2.3)
**for** *tuple* $(s, a)$ **do**

    $y \sim Z(s, a)|_{\tau > \eta}$
    $\xi(s, a), \sigma(s, a) = \text{MLE}[\text{GPD}(y - Z(s, a)|_{\tau=\eta}]$

**Policy Improvement:** (Section 7.3)
Update policy $\pi$ according to Eqn.12

---

We provide an algorithmic flow of EVAC in Algorithm 1. The EVAC algorithm receives the critic and actor network parameter initializations, the GPD parameter initializations and the threshold quantile level $\eta$ as inputs. Firstly, any tuple $(s, a, r, s, a' = \pi(s'))$ or equivalently a batch of such tuples is considered for updating the state action value distribution. To accomplish this, the excess over the threshold quantile level $\eta$, is computed by sampling the GPD distribution i.e. $x \sim GPD\big(\xi(s', a'), \sigma(s', a')\big)$. Next, two kind of samples are obtained. One from the non-tail region $(Z'_L)$ and another from the tail region $(Z'_H)$. Then these samples are selected in proportions $\eta$ and $(1 - \eta)$ respectively to update the critic through standard quantile regression performed on the entire batch.

Secondly, a tuple of state action pairs $(s, a)$ or equivalently a batch of them is considered to update the GPD parameters. The updated state action value distribution is sampled to obtain the the excess over the threshold quantile for all elements in the batch. The parameters of the GPD distribution are then updated using maximum likelihood estimation on the excess samples.

Finally, the policy is updated using CVaR optimization on the updated state action value distribution.

## 8 Experimental Evaluation

As detailed earlier, extreme rare risk scenarios can have catastrophic outcomes. Although it is very pertinent to use real world complex control environments to study risk aversion, such complex scenarios often lack

open source simulators. As a result, we resort to creating such extreme rare risk scenarios in open source environments.

Thus, in order to evaluate rare risk aversion of each agent, we define special environments following the work of Ma et al. (2021); Urpí et al. (2021) where risk scenarios are simulated. However, to simulate extreme risk scenarios we introduce some modifications.

### 8.1 Generating extreme risk scenarios

In order to replicate scenarios where the state-action value distribution $Z^\pi(s, a)$ encompasses exceedingly rare extreme values with minimal probabilities, we deliberately design rewards $\mathbf{r}(\mathbf{s}, \mathbf{a})$ that can take very low values (representing catastrophic events) with extremely low probabilities. This mirrors realistic situations where the reward for a given state-action pair is drawn from a distribution that seldom yields low values, corresponding to high penalties.

To illustrate a strategy simulating low-probability, high-penalty events, we incorporate a penalty term into the reward with a low probability. Essentially, this penalty term is influenced by a Bernoulli random variable parameterized by $\mathbf{p}$, offering control over the rarity of penalizing events. The parameter $\mathbf{p}$ determines the infrequency of penalties.

We experiment on two benchmark Open-AI environments Brockman et al. (2016) namely Mujoco environments and Safety-gym environments Ji et al. (2023).

### 8.2 Mujoco Environments

For creating rare risky events in the Mujoco Environments, we modify the reward using a wrapper function which penalizes the reward for certain state action pairs with a certain probability (which is typically small to simulate rare risky events). Our reward penalization setup is similar to the set up of Urpí et al. (2021). We primarily experiment with three Mujoco environments namely, HalfCheetah, Hopper and Walker2d environments. Denoting $\mathbf{R}(\mathbf{s}, \mathbf{a})$ as the original non-penalized reward, we define the stochastic penalized reward $\mathbf{r}(\mathbf{s}, \mathbf{a})$. For the Half-Cheetah environment,

$$\mathbf{r}(\mathbf{s}, \mathbf{a}) = \mathbf{R}(\mathbf{s}, \mathbf{a}) + \mathbb{I}_{\mathbf{v} > \alpha} \mathbf{L} \cdot \mathcal{B}_{\mathbf{p}}$$

where $\mathbb{I}$ is the indicator function, $\mathbf{v}$ is the velocity of the HalfCheetah agent, $\mathbf{L} = -50$, is a penalization weight, $\alpha = 2.5$ is the threshold velocity of the HalfCheetah agent above which the agent gets penalized rarely. The sample of the Bernoulli random variable is denoted by $\mathcal{B}_p$. To simulate extreme rare penalties, the Bernoulli distribution parameter $\mathbf{p}$ is made equal to 0.05 i.e. ($\mathbf{p} = 0.05$), which indicates the frequency at which the state action pair $(s, a)$ is penalized.

For the Hopper environment,

$$\mathbf{r}(\mathbf{s}, \mathbf{a}) = \mathbf{R}(\mathbf{s}, \mathbf{a}) + \mathbb{I}_{|\theta| > \alpha} \mathbf{L} \cdot \mathcal{B}_{\mathbf{p}}$$

where $\mathbb{I}$ is the indicator function, $|\theta|$ denotes the angle of Hopper, $\mathbf{L} = -50$ is a penalization weight, for the reward, $\alpha = 0.03$ is the threshold angle over which the agent gets penalized. The Bernoulli distribution parameter is set to $\mathbf{p} = 0.03$.

For the Walker2d environment,

$$\mathbf{r}(\mathbf{s}, \mathbf{a}) = \mathbf{R}(\mathbf{s}, \mathbf{a}) + \mathbb{I}_{|\theta| > \alpha} \mathbf{L} \cdot \mathcal{B}_{\mathbf{p}}$$

where $\mathbb{I}$ is the indicator function, $|\theta|$ denotes the angle of the Walker2d agent, $\mathbf{L} = -30$ is a penalization weight for the reward, $\alpha = 0.2$ is the threshold angle over which the agent gets penalized. The Bernoulli distribution parameter $\mathbf{p} = 0.03$.

For all Mujoco environments, we analyze the CVaR (in Eqn.12) and its variance. Next, in order to test for risk aversion of the agent, we define a metric namely the percentage failure which is defined as :

$$\text{Percentage Failure} = \mathrm{E_N}[\mathbb{I}_{q > \alpha}] * 100$$

where $E_N$ represents the empirical mean over $N$ episodes, $q$ is the quantity of interest, i.e. the velocity $\mathbf{v}$ for the Half-Cheetah and the angle $|\theta|$ for the Hopper and Walker.

The percentage failure is indicative of the fraction of the times in an episode that the agent enters into state-action pairs that are penalized rarely. The percentage of failure naturally quantifies the risk aversion ability of the agent in question. Additionally, we also include the cumulative reward collected in each episode to assess the performance of the agent in addition to its risk aversion capability.

### 8.3 Safety-gym Environments

We also perform extensive experiments on another suite of environments, namely the safety-gym benchmark Ji et al. (2023). Safety-gym environments consist of configurable robots with programmable reward functions. We use the 'Point' robot and the 'goal' task. Thus, in our setup a particular robot (the 'point' robot) is tasked with reaching a goal location on the arena. The robot receives a reward with respect to distance from the goal.

We follow the method of reward penalization used by Ma et al. (2021) to introduce rare risky rewards. We define certain circular regions on the arena, termed 'hazards' which rarely lead to penalizations. To introduce extreme rare risks, the nature of reward penalization is

$$\mathbf{r(s, a) = R(s, a)} + \mathbb{I}_{\mathbf{q} \in \mathcal{H}} \mathbf{L} \cdot \mathcal{B}_{\mathbf{p}}$$

where $\mathbf{R(s, a)}$ is the original non-penalized reward, $\mathbf{r(s, a)}$ is the stochastic penalized reward, $\mathbf{q}$ denotes the current location of the robot in the arena, $\mathcal{H}$ includes positions of all the hazards in the arena. The sample of the Bernoulli random variable is denoted by $\mathcal{B}_p$ and $\mathbf{p}$ is the Bernoulli frequency of penalization. The penalty weight $\mathbf{L} = -10$. Thus, the agent suffers a rare penalty when in the hazard region. We create two scenarios for extreme risks with respect to the placement of the hazard and the size (radius) of the hazard.

**Safety-gym Scenario-A:** In this experimental setup, there is a single hazard of large radius placed along the straight line path (shortest distance) between the start position and the goal position. The agent needs to learn to avoid the large hazard region. We set the Bernoulli penalization rate $\mathbf{p} = 0.05$ for Scenario-A.

**Safety-gym Scenario-B:** In this more challenging experimental setup, there are multiple hazards of smaller radius placed closer to the goal. The agent needs to discover the optimal path to avoid all hazards and reach the goal. We make the setup even more challenging by setting the penalization parameter to $\mathbf{p} = 0.03$ to introduce more extreme risks.

We provide visual representations of Scenario-A and Scenario-B in Figure 5 and Figure 6 respectively in Section H of the Appendix.

### 8.4 Baselines

We compare EVAC over different baseline algorithms such as DDPG-RAAC Urpí et al. (2021), TD3-RAAC (TD3 version of DDPG-RAAC) and DSAC (with single critic) Ma et al. (2020). RAAC uses quantile regression to construct $Z^\pi(s, a)$ and uses the CVaR to optimize the policy. DSAC additionally maximizes the entropy of the policy to ensure optimal exploration. The difference between our approach (EVAC) and the compared baselines is that we additionally approximately model the tail of the state-action value distribution using the shifted and scaled GPD distribution.

### 8.5 Result Discussion

As can be seen in Table 1, the EVAC agents exhibit good risk averse behavior by avoiding risky reward state actions. The percentage failure for the EVAC agents is noticeably smaller when subject to even extreme risk scenarios.

Secondly, the CVaR of the EVAC agents is higher than the baseline methods. Additionally, the standard deviation of the CVaR across different runs for the EVAC agent is smaller in comparison to other baselines.

| Environment | Algorithm | Percentage Failure | Cumulative Reward | CVaR |
|---|---|---|---|---|
| HalfCheetah | RAAC-DDPG | 16.55± 4.43 | 637.81± 319.78 | 135.18 ± 13.02 |
| | RAAC-TD3 | 41.3± 16.6 | 836.8± 195.85 | 129.81 ± 38.75 |
| | D-SAC | 30.04 ± 22.26 | 1356.05 ± 269.63 | 149.76 ± 27.38 |
| | EVAC | **2.87 ± 1.3** | **1502.46 ± 94.25** | **156.71 ± 11.07** |
| Hopper | RAAC-DDPG | 66.94 ± 11.79 | 410.62 ± 315.12 | 27.58 ± 34.41 |
| | RAAC-TD3 | 48.78 ± 12.36 | 718.53 ± 355.55 | 27.87 ± 65.35 |
| | D-SAC | 57.34 ± 11.42 | 782.85 ± 400.78 | 51.5 ± 95.93 |
| | EVAC | **2.09 ± 4.18** | **875.89 ± 299.75** | **76.61 ± 28.15** |
| Walker2d | RAAC-DDPG | 68.05 ± 11.62 | 123.17 ± 66.46 | 1.31 ± 6.6 |
| | RAAC-TD3 | 22.18 ± 13.02 | 308.66 ± 134.99 | -9.52 ± 14.83 |
| | D-SAC | 24.75 ± 11.33 | 598.5 ± 128.38 | 3.76 ± 8.34 |
| | EVAC | **4.98 ± 7.23** | **668.18 ± 463.68** | **4.37 ± 3.22** |

Table 1: Performance metrics on Mujoco environments under a penalization rate of $\mathbf{p} = 0.05$ for the Half Cheetah environment and $\mathbf{p} = 0.03$ for the Hopper and Walker2d environments. We record CVaR at $x_1 = 0.95$ (0.05 if reward is not negated) and threshold quantile level $\eta = 0.96$. Inference is done on 5 trained agents. Each trained agent completes an episode in inference mode acting on the learnt policy. The results are expressed as mean ± standard deviation across 5 trained agents.

.

| Environment | Algorithm | Percentage Failure | Cumulative Reward | CVaR |
|---|---|---|---|---|
| Scenario-A | RAAC-DDPG | 14.03 ± 11.4 | 5.41 ± 0.24 | 1.59 ± 0.09 |
| | RAAC-TD3 | 12.13± 5.94 | 5.55± 0.23 | 1.39 ± 0.13 |
| | D-SAC | 22.31 ± 13.67 | 2.13 ± 4.07 | 1.33 ± 0.18 |
| | EVAC | **0.0 ± 0.0** | **5.71 ± 0.0** | **1.7 ± 0.08** |
| Scenario-B | RAAC-DDPG | 20.2± 14.66 | 5.22± 0.41 | 1.58 ± 0.1 |
| | RAAC-TD3 | 22.34± 9.93 | 4.87± 0.47 | 0.64 ± 0.41 |
| | D-SAC | 15.47 ± 4.84 | 5.1 ± 0.24 | 1.56 ± 0.09 |
| | EVAC | **2.47 ± 3.9** | **5.61 ± 0.12** | **2.0 ± 0.06** |

Table 2: Performance metrics on the Safety-gym environment under a penalization rate of $\mathbf{p} = 0.05$ for Scenario-A and $\mathbf{p} = 0.03$ for Scenario-B. We record CVaR at $x_1 = 0.95$ (0.05 if reward is not negated). We set the threshold quantile $\eta = 0.96$. The results are expressed as mean ± standard deviation across 5 trained agents.

This is in concurrence to the analysis in Section 6.1. The lower variance implies that the EVAC agent is able to consistently and precisely quantify risk in the tail region of the state action value distribution and thereby avoid risky state action pairs (which is demonstrated by the low percentage failures).

The cumulative rewards in Table 1 show higher performances than other baselines. In the Mujoco environments, the cumulative reward of DSAC is slightly smaller to EVAC and comes at the cost of entering rare risky regions. We observe that RAAC agents are not as risk averse and do not improve cumulative performance either in extreme rare risk scenarios. EVAC agents while achieving higher cumulative performance, exhibit very good risk aversion too. This implies that the EVAC agents explore systematically within the 'legal' and non-risky regions to maximize cumulative reward. We observe similar trends in risk aversion and the CVaR in the safety-gym environments in Table 2. The cumulative rewards collected by EVAC agents is the maximum in case of the safety-gym environments which demonstrates the EVAC agents' ability to recognize and avoid extremely rare risk hazards (at even 3% extreme penalty rate) while trying to maximize cumulative reward.

In summary, the obtained CVaR values demonstrate that EVAC agents improve optimization of the risk sensitive objective. The higher cumulative rewards indicate that EVAC agents explore well and are not too conservative. The near zero failure percentages demonstrate that the EVAC agents can be risk averse even under extremely rare penalizations.

We perform several ablation studies in the Appendix. We perform ablation studies on varying the penalization rate $\mathbf{p}$ to assess the ability of EVAC to adjust to varying levels of extreme risks in Section D. We also discuss the sensitivity of the EVAC algorithm on the threshold quantile level $\eta$ in Section F. The CVaR quantile $x_1$, controls risk averse behavior. We perform ablation studies with stricter $x_1$ values on all baseline algorithms in Section G to ascertain if simply changing the $x_1$ value leads to conservative behavior. We find that simply increasing $x_1$ does not lead to risk aversion in extreme risk scenarios and only additional tail modelling aids in risk aversion. We perform a comparative analysis between the estimated tail distribution of the RL agents against the ground truth distribution for EVAC and other baselines in Section E. We describe the experimental setup and hyperparameters in greater detail in Section C. In Section H we provide visual illustrations of the risk averse trajectories of the EVAC agents on the Safety-gym environments. We also provide a proof of the convergence of the Bellman operator used in EVAC in Section A for a fixed policy. We provide details of the MLE estimation procedure in Section B of the Appendix.

## Acknowledgement

This research was supported by the U.S. Department of Energy, through the Offce of Advanced Scientifc Computing Research's "Data-Driven Decision Control for Complex Systems (DnC2S)" project. The research participation of the University of California at Santa Barbara was supported under award number DE-SC0021321. Pacifc Northwest National Laboratory is operated by Battelle Memorial Institute for the U.S. Department of Energy under Contract No. DEAC05-76RL01830. Oak Ridge National Laboratory is operated by UT-Battelle LLC for the U.S. Department of Energy under contract number DE-AC05-00OR22725. The Jefferson Science Associates (JSA) operates the Thomas Jefferson National Accelerator Facility for the U.S. Department of Energy under Contract No. DE-AC05-06OR23177.

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

# A    Proof of convergence of the Bellman Operator in EVAC

In this section, we set out to prove the convergence of the Bellman update Eqn.11.

$$T^\pi Z(s,a) = r(s,a) + \gamma\Big[Z_L(s',a') + (1-\eta)Z_H(s',a')\Big]$$

***Definition 1:*** For any two random variables $J_1(s,a)$ and $J_2(s,a)$ with distributions $Z_1(s,a)$ and $Z_2(s,a)$ with inverse CDF functions $F^{-1}_{J_1(s,a)}$ and $F^{-1}_{J_2(s,a)}$ respectively, the Wasserstein distance $d_p$ is defined as:

$$d_p\Big(F_{J_1(s,a)}, F_{J_2(s,a)}\Big) = \Big(\int_0^1 |F^{-1}_{J_1(s,a)}(u) - F^{-1}_{J_2(s,a)}(u)|^p du\Big)^{1/p}$$

Equivalently, the maximal Wasserstein distance $\bar{d}_p$ is defined as:

$$\bar{d}_p(F_{J_1}, F_{J_2}) = \sup_{s,a} d_p\Big(F_{J_1(s,a)}, F_{J_2(s,a)}\Big)$$

***Property 1:*** **For a scalar constant $r$, the shifted random variables $J_1(s,a)+r$ and $J_2(s,a)+r$ have**

$$d_p\Big(F_{J_1(s,a)+r}, F_{J_2(s,a)+r}\Big) = d_p\Big(F_{J_1(s,a)}, F_{J_2(s,a)}\Big)$$

***Property 2:*** **For a real constant scaling factor $0 < \gamma < 1$, the scaled random variables $\gamma J_1(s,a)$ and $\gamma J_2(s,a)$ have**

$$d_p\Big(F_{\gamma J_1(s,a)}, F_{\gamma J_2(s,a)}\Big) \leq \gamma d_p\Big(F_{J_1(s,a)}, F_{J_2(s,a)}\Big)$$

***Definition 1:*** For a distribution Z, a quantile level $\eta$ and its corresponding quantile $Z_\eta$, we define the non-tail distribution $Z_L = Pr(Z \leq Z_\eta)$ and the non-tail distribution $Z_H = \frac{1}{1-\eta}Pr(Z > Z_\eta)$.

***Theorem 1:*** **Let $\mathcal{Z}$ denote the space of all state action value distributions. For the state action value distribution $Z(s,a) = Z_L(s,a) + (1-\eta)Z_H(s,a)$ , where $Z_L$ represents the non-tail region of $Z$ and $Z_H$ represents the tail region of $Z$ (as described in Definition 1), the Bellman operator $T^\pi : \mathcal{Z} \times \mathcal{Z}$, is a $\gamma$ contraction under the maximal Wasserstein distance metric $\bar{d}_p$.**

**Note:** For notational convenience, we express $d_p\Big(F_{J_1(s,a)}, F_{J_2(s,a)}\Big)$ as $d_p\Big(Z_1(s,a), Z_2(s,a)\Big)$.

**Proof:**

$$
\begin{aligned}
&d_p(T^\pi Z_1, T^\pi Z_2) \\
&= d_p\Big(r(s,a) + \gamma\Big[Z_{L_1}(s',a') + (1-\eta)Z_{H_1}(s',a')\Big], r(s,a) + \gamma\Big[Z_{L_2}(s',a') + (1-\eta)Z_{H_2}(s',a')\Big]\Big) \\
&= d_p\Big(\gamma\Big[Z_{L_1}(s',a') + (1-\eta)Z_{H_1}(s',a')\Big], \gamma\Big[Z_{L_2}(s',a') + (1-\eta)Z_{H_2}(s',a')\Big]\Big) \\
&\leq \gamma d_p\Big(\Big[Z_{L_1}(s',a') + (1-\eta)Z_{H_1}(s',a')\Big], \Big[Z_{L_2}(s',a') + (1-\eta)Z_{H_2}(s',a')\Big]\Big) \\
&= \gamma d_p\Big(Z_1(s',a'), Z_2(s',a')\Big)
\end{aligned}
$$

$$
\begin{aligned}
\bar{d}_p(T^\pi Z_1, T^\pi Z_2) &= \sup_{s,a} d_p\Big(T^\pi Z_1(s,a), T^\pi Z_2(s,a)\Big) \\
&\leq \gamma \sup_{s,a} d_p\Big(Z_1(s,a), Z_2(s,a)\Big) \\
&= \gamma \bar{d}_p\Big(Z_1(s,a), Z_2(s,a)\Big)
\end{aligned}
$$

From the above equation, we prove that the $T^\pi$ operator is a contraction and that the Bellman update with the GPD tail distribution converges.

## B    MLE Estimation of the parameters of the GPD Distribution

The CDF of the GPD distribution $F_{\xi,\sigma}(x)$ is given by:

$$
\begin{cases}
1 - \left(1 + \frac{\xi x}{\sigma}\right)^{-1/\xi} \text{ for } \xi \neq 0 \\
1 - \exp(-\frac{x}{\sigma}) \text{ for } \xi = 0
\end{cases}
$$

The log-density function (log-PDF) of the same GPD distribution is given by:

$$
\log f_{\xi,\sigma}(x) = \begin{cases}
-\log(\sigma) + \left(-1/\xi - 1\right) \log\left(1 + \xi x/\sigma\right) \text{ for } \xi \neq 0 \\
-\log(\sigma) - x/\sigma \text{ for } \xi = 0
\end{cases}
\tag{13}
$$

$$
\frac{\partial \log f_{\xi,\sigma}(x)}{\partial \xi} = \left(-1/\xi - 1\right)\left(\frac{1}{1 + \xi x/\sigma}\right) \cdot \frac{x}{\sigma} + \log\left(1 + \xi x/\sigma\right)\left(1/\xi^2\right) = 0
$$

$$
\frac{\partial \log f_{\xi,\sigma}(x)}{\partial \sigma} = -\frac{1}{\sigma} + \left(-1/\xi - 1\right) \cdot \frac{1}{1 + \xi x/\sigma} \cdot \xi x = 0
$$

However when $\xi = 0$; we are left with the MLE estimation of the parameter $\sigma$ of the exponential distribution

$$
\frac{\partial \log f_{0,\sigma}(x)}{\partial \sigma} = -\frac{1}{\sigma} + \frac{x}{\sigma^2} = 0
\tag{14}
$$

We make the GPD parameter $\sigma(s,a), \xi(s,a)$ be parameterized and represent it by $\sigma_\theta(s,a), \xi_\phi(s,a)$. Given a batch of transition tuples $(s_i, a_i, r_i, s_i')$ ; $i = 1 \rightarrow B$, where $B$ is the batch size, we source samples from the GPD distribution by sampling $K$ samples $x_k$ from $Z_H^\pi(s,a)$, the constructed GPD distribution. We then compute the empirical log-likelihood loss $\mathcal{L}$ that needs to be maximized.

$$
\mathcal{L}_\phi = \frac{1}{B} \sum_{i=1}^{B} \cdot \frac{1}{K} \sum_{k=1}^{K} \left[\frac{\partial \log f_{\xi,\sigma}(x_k)}{\partial \xi_\phi}\right]
\tag{15}
$$

$$
\mathcal{L}_\theta = \frac{1}{B} \sum_{i=1}^{B} \cdot \frac{1}{K} \sum_{k=1}^{K} \left[\frac{\partial \log f_{\xi,\sigma}(x_k)}{\partial \sigma_\theta}\right]
\tag{16}
$$

where $x_k \sim Z_H^\pi(s,a)$, $K$ is the number of samples sampled from the GPD distribution $Z_H^\pi(s,a)$. We set $K = 100$ and $B = 128$ for all experimentation. We perform gradient ascent on the parameters $\theta$ by using the empirical loss:

$$
\begin{aligned}
\theta &: \theta + \alpha_{lr} \cdot \mathcal{L}_\theta \\
\phi &: \phi + \alpha_{lr} \cdot \mathcal{L}_\phi
\end{aligned}
\tag{17}
$$

where $\alpha_{lr}$ is the learning rate. Thus, there is a computation overhead of $\mathcal{O}(B \cdot K)$ in updating the parameters of the GPD distribution.

## C    Experimental details and Hyperparameter setting

We use an actor critic framework , where the critic is a quantile critic. Both the actor and critic have 3 layers with hidden size being 128.

## C.1   Mujoco Environments

During training and inference, the max episode length of the agent is set to 1000. During training, the agents were trained for 100,000 time steps on the whole. The batch size $B = 128$ and we set $K$, the number of samples sampled from the GPD distribution to 50. We set the learning rates for the actor and critic to 0.001 in all cases. The discount factor $\gamma = 0.99$ for all cases too. The soft update parameter $\tau = 0.02$ for all our experiments on the Hopper and Walker2d, while $\tau = 0.01$ for the HalfCheetah environment.

## C.2   Safety Gym Environments

During training and inference, the max episode length of the agent is set to 1000. During training, the agents were trained for 70,000 time steps on the whole for the safety-gym suite.

In all of the ablation studies, we report our results on 3 different trained agents.

## D   Risk Awareness and Performance with changing Penalization rate p

We recall that the penalization rate $\mathbf{p}$ is defined as the Bernoulli distribution parameter that is used to create rare risky rewards of the form $\mathbf{r(s, a) = R(s, a) + \mathbb{I}_{v > \alpha}L.\mathcal{B}_p}$, for the Half-Cheetah environment. As $\mathbf{p}$ becomes smaller, the penalty becomes rarer, thereby indicating more extreme risk scenarios.

In this section we change this penalization rate $\mathbf{p}$ of the reward for the HalfCheetah-v3 environment. We move from an extreme rare reward penalty of $p = 0.03$ to a fairly frequent penalty rate of $p = 0.1$. For our experimentation purposes, we fix the threshold quantile $\eta = 0.96$ for both EVAC and RAAC-TD3. From Table 3 however, as the penalization rate $p$ becomes smaller (rare risky events), the EVAC agent still exhibits very small percentages of failure while the cumulative reward is still very high. This bolsters the fact that the EVAC agent still explores reasonably while being risk averse.

| Algorithm | $p$ | Percentage Failure | Cum. Reward | CVaR |
|-----------|-----|--------------------|-----------|------|
| EVAC | 0.03 | **6.43 $\pm$ 4.99** | 1412.81 $\pm$ 292.22 | 140.66 $\pm$ 23.57 |
| | 0.05 | **2.87 $\pm$ 1.3** | 1502.46 $\pm$ 94.25 | 156.71 $\pm$ 11.07 |
| | 0.1 | **1.07 $\pm$ 0.63** | 1160.8 $\pm$ 264.46 | 111.48 $\pm$ 33.65 |
| RAAC-TD3 | 0.03 | 30.23 $\pm$ 12.21 | 847.86 $\pm$ 145.21 | 85.66 $\pm$ 24.12 |
| | 0.05 | 41.3$\pm$ 16.6 | 836.8$\pm$ 195.85 | 129.81 $\pm$ 38.75 |
| | 0.1 | 13.97 $\pm$ 5.58 | 366.0 $\pm$ 453.85 | 104.92 $\pm$ 7.08 |

Table 3: Table showing various performance metrics as the penalization rate $p$ is varied for the HalfCheetah environment (with fixed threshold quantile $\eta = 0.96$).

.

## E   Comparison of the modelled distribution with the Ground-truth distribution

Since we believe that the use of EVT theory helps in accurately modelling the tail of the distribution of the future sum of discounted returns, we compare the modelled distribution $Z^\pi(s, a)$ of every algorithm (baselines and EVAC) against the true empirically obtained distribution of the sum of future discounted returns.

We freeze the initial state to be the same across all algorithms (RAAC-DDPG, RAAC-TD3, D-SAC, EVAC) and collect trajectories according to optimal policy defined for each algorithm. For example, each algorithm has a defined optimal policy and a trajectory $\mathcal{T}^t = \{R_1, R_2, \cdots\}$ is collected by following the optimal policy of that algorithm. For such a trajectory $\mathcal{T}^t$, we compute the future discounted sum of rewards $J^t = \sum_i \gamma^i R_i$ for the very first state in the trajectory. We repeat this process of trajectory collection for the same initial state $N$ times to estimate the distribution of the sum of future discounted returns for each algorithm from $\{J^t\}_{t=1}^N$.

On the other hand we sample values from the quantile critic of each algorithm to obtain the modelled distribution $Z^\pi(s, a)$. We collect $\{z^j\}_{j=1}^M$, where $z^j \sim Z(s, \pi(s))$. It is to be noted that since each algorithm

has a different optimal policy, the empirical distribution of the sum of future discounted returns would vary for each algorithm.

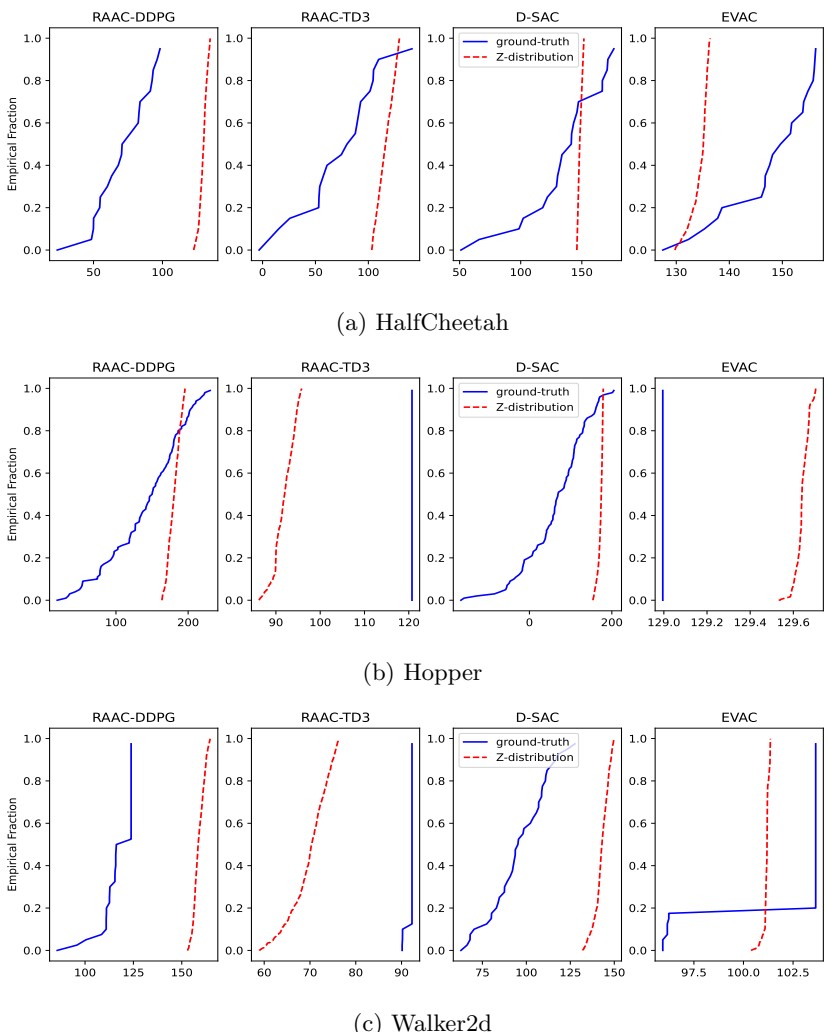

(a) HalfCheetah

(b) Hopper

(c) Walker2d

Figure 4: Comparison of the empirical CDF of the modelled distribution (in dashed red) and the empirical CDF of the ground truth (in solid blue) sum of discounted rewards. All plots are plotted for non-negated rewards. The tail behavior (left tail) of the EVAC agents almost precisely matches the true empirical distribution of the sum of the future discounted rewards. (Note that the scale of the X-axis is different for each algorithm and environment)

We then plot the empirical CDF of both $\{J^t\}_{t=1}^N$ (which is called the ground truth) and $\{z^j\}_{j=1}^M$ (modelled Z-distribution). These empirical CDFs are plotted for each algorithm and for each Mujoco agent in **Figure 4**.

We compute the 1-Wasserstein distance between the ground truth and the modelled distribution in Table 4. The 1-Wasserstein distance $d_1(X, Y)$ between two random distributions $X$ and $Y$ is defined as:

$$d_1(X, Y) = \Big( \int_0^1 |F_X^{-1}(u) - F_Y^{-1}(u)| du \Big)$$

Firstly for the HalfCheetah, we see that the left tail of the ground truth and the modelled distribution for EVAC, both concentrate in the same region of around 100-120. For the Hopper, the ground truth and the modelled Z distribution both are almost perfectly aligned at a value of around 129 to 130. For the Walker2d,

| Environment | Algorithm | 1-Wasserstein distance (↓) |
|---|---|---|
| HalfCheetah | RAAC-DDPG | 55.05 |
| | RAAC-TD3 | 57.69 |
| | D-SAC | 37.55 |
| | EVAC | **14.89** |
| Hopper | RAAC-DDPG | 38.22 |
| | RAAC-TD3 | 28.77 |
| | D-SAC | 86.87 |
| | EVAC | **0.64** |
| Walker2d | RAAC-DDPG | 42.70 |
| | RAAC-TD3 | 21.95 |
| | D-SAC | 47.40 |
| | EVAC | **3.32** |

Table 4: Table showing 1-Wasserstein distance between the empirical ground truth distribution of the sum of discounted rewards and the modelled distribution as shown in Figure 4

we see a similar trend in the tails of the groundtruth and modelled distribution for the EVAC algorithm. (Note that the scale of the X-axis is different for each algorithm and environment). From Table 4 and Figure 4, we notice that there is generally a smaller discrepancy between the ground-truth distribution of the EVAC agent and its corresponding modelled distribution, in comparison with other baselines that do not use EVT to model the tail behavior. Thus, EVAC agents are able to better model the true underlying distribution, especially the tail regions.

## F   Risk Awareness and Performance with changing threshold quantile parameter $\eta$

We next address the sensitivity to risk awareness when the threshold quantile parameter $\eta$ is changed for the HalfCheetah-v3 environment. The value of $\eta$ indicates the threshold quantile level after which the GPD approximation for $Z_H^\pi(s, a)$ is made.

We fix the penalization rate $\mathbf{p} = 0.05$ while changing $\eta$ continuously. We also fix $x_1 = 0.95$ and $x_2 = 1.0$ for CVaR calculation.

| Environment | $\eta$ | Percentage Failure | Cum. Reward | CVaR |
|---|---|---|---|---|
| HalfCheetah | 0.90 | $35.17 \pm 23.26$ | $946.65 \pm 234.75$ | $183.68 \pm 35.38$ |
| | 0.96 | $\mathbf{2.87 \pm 1.3}$ | $1502.46 \pm 94.25$ | $156.71 \pm 11.07$ |

Table 5: Table illustrating various performance metrics as the threshold quantile $\eta$ is varied for the HalfCheetah environment (with fixed penalization rate $p = 0.05$).

.

We observe that the choice of the threshold quantile $\eta$ plays an important role in risk mitigation too. We observe that risk sensitivity increases with increase in the value of $\eta$. We posit that this is due to better approximation of the GPD distribution as $\eta$ increases (Theorem 4.2). Table 5, as $\eta$ increases, the EVAC agents become very risk averse while producing good cumulative rewards. This underscores the effectiveness of using EVT theory for risk mitigation in reinforcement learning.

## G   Effect of increasing $x_1$ in CVaR optimization (more conservative behavior)

We recall from Equation 12 in the manuscript that $x_1$ is used to control the level of risk aversion. We re-write Equation 12 from the manuscript below.

$$\pi^* = \arg\min_{\pi} \mathrm{CVaR}(x_1)$$

$$= \arg\min_{\pi} \frac{1}{x_2 - x_1} \int_{\tau=x_1}^{x_2=1} Z\Big(s, \pi(s)\Big)\Big|_{\tau}$$

As $x_1 \to 1$, the policy is incentivized to optimize the right tail and is thus made more risk averse. Instead of $x_1 = 0.95$, we set $x_1 = 0.99$ to ensure the highest level of conservatism.

| Environment | Algorithm | Percentage Failure | Cum. Reward | CVaR |
|---|---|---|---|---|
| | RAAC-DDPG | $47.07 \pm 17.64$ | $677.15 \pm 173.03$ | $126.22 \pm 115.43$ |
| HalfCheetah | RAAC-TD3 | $30.47 \pm 4.97$ | $360.55 \pm 242.4$ | $104.21 \pm 12.0$ |
| | D-SAC | $35.7 \pm 40.42$ | $1089.13 \pm 395.36$ | $143.37 \pm 50.41$ |
| | EVAC | $\mathbf{1.03 \pm 1.26}$ | $872.62 \pm 171.62$ | $151.45 \pm 4.33$ |

Table 6: Table showing the the percentage failure and also the cumulative reward as risk sensitivity parameter $x_1$ is made stricter at 0.99 instead of 0.95 for the HalfCheetah environment (with fixed penalization rate $p = 0.05$ and threshold quantile $\eta=0.96$).

.

From Table 6, even when the CVaR objective is made very risk averse (at $x_1 = 0.99$), the baseline agents still do not demonstrate good risk mitigation. The vanilla quantile based methods which depend on sampling exclusively, may not model the tail behavior accurately enough. This implies that even if $x_1$ is made arbitrarily large (more risk averse) in such agents, proper risk mitigation may still not be observed. EVAC on the other hand demonstrates good risk aversion. This further acts as a motivation to using EVAC-like agents which seem to evidently show better risk mitigation.

## H  Trajectory Visualization on Safety-gym environments

In this section, we provide visual depictions of the trajectories followed by different agents in the Safety-gym environment. We represent the hazards by red circles. The starting position is indicated in green and the goal position is indicated in blue in Figures 5 and 6. As discussed in the manuscript, the goal of the agent is to navigate to the goal while avoiding the red 'hazard circles'. In the two scenarios A and B, we observe that the EVAC agent is able to efficiently learn the locations of the hazard and avoid such hazards at all times. However, due to the extremely rare penalty rate of $\mathbf{p} = 0.05$ for Scenario-A and $\mathbf{p} = 0.03$ for Scenario-B, the other baseline agents, do not seem to recognize the low probability events and often enter the hazardous blue circles.

## I  Histograms showcasing risk avoidance

We illustrate the histogram of the quantities of interest in the Mujoco environments in Figure 7. We perform inference on the trained agents and record the quantities of interest for each Mujoco environment. We observe that the quantity of interest (i.e. velocity of the HalfCheetah and the angle of the Hopper) are mostly within the safe region as anticipated.

## J  Experiments on Discrete Action Space

To extend the EVAC algorithm to discrete action spaces we experiment on an autonomous driving environment (Leurent, 2018) where the agent is tasked with driving a car on a highway. The agent is rewarded for moving fast and avoiding any collisions. However, the car agent may get penalized rarely by highway patrol for speeding over the speed limit. The goal of the agent is thereby to navigate on the highway at the speed limit while avoiding any collisions with other cars on the highway. To make speeding penalties rare, we ensure that the highway patrol is present only 5% of the times.

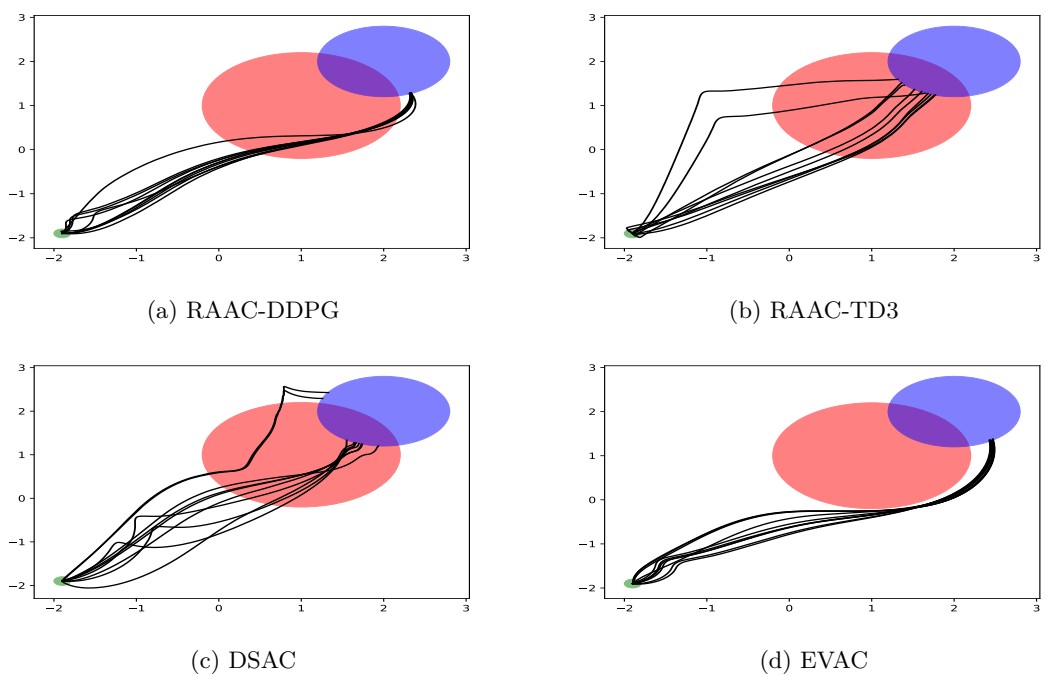

(a) RAAC-DDPG

(b) RAAC-TD3

(c) DSAC

(d) EVAC

Figure 5: Trajectories visualized for the Safety-gym Scenario -A with $\mathbf{p} = 0.05$. The EVAC agent successfully navigates to the goal by recognizing the blue hazards.

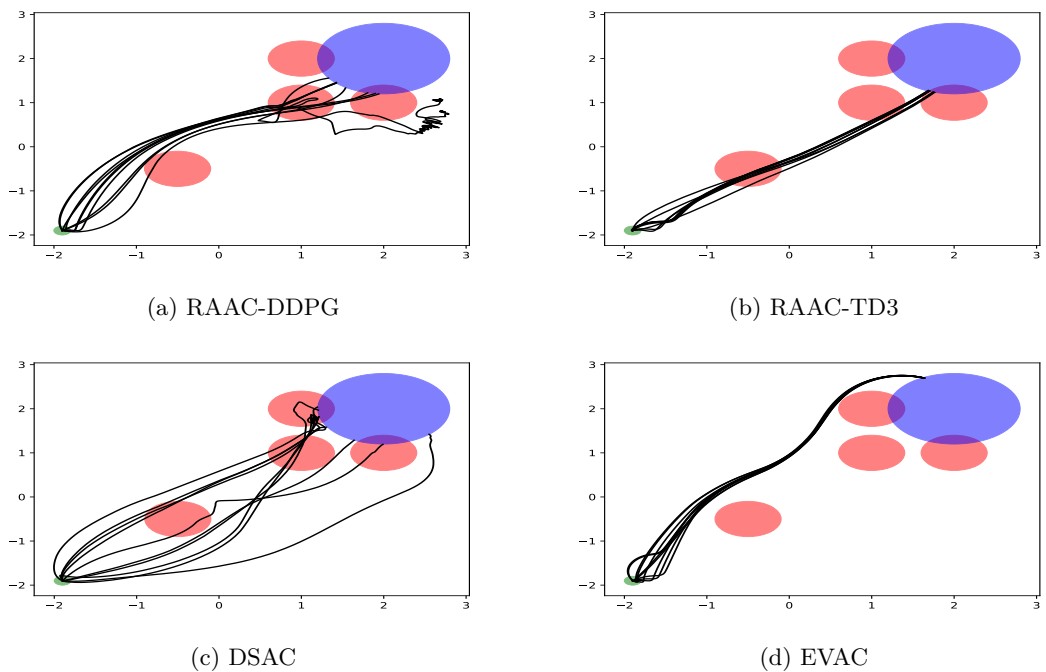

(a) RAAC-DDPG

(b) RAAC-TD3

(c) DSAC

(d) EVAC

Figure 6: Trajectories visualized for the Safety-gym Scenario-B with $\mathbf{p} = 0.03$. The EVAC agent successfully navigates to the goal by recognizing the blue hazards.

The highway environment has a continuous state space with dimension 25. It has a discrete action space with 5 actions. We illustrate the highway environment below in Figure 8.

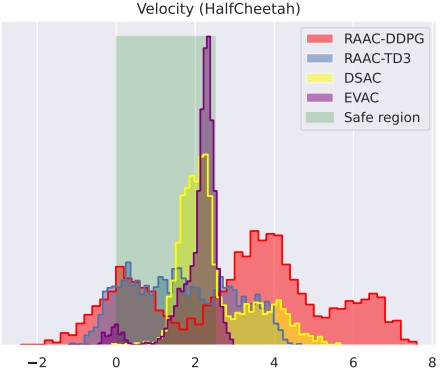
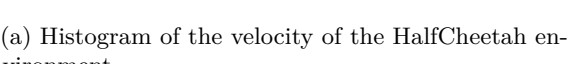

(a) Histogram of the velocity of the HalfCheetah environment

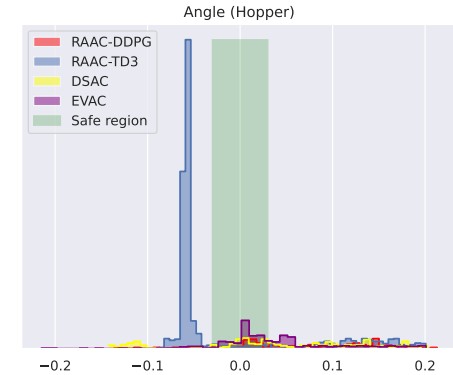

(b) Histogram of the angle of the Hopper environment

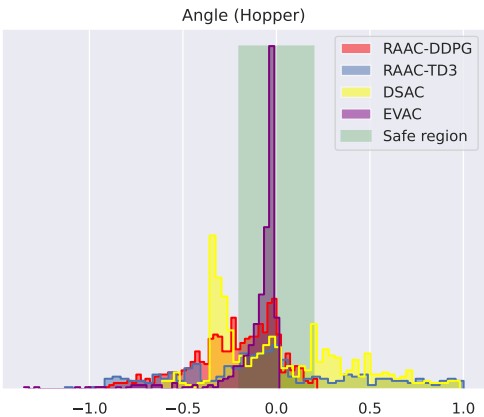

(c) Histogram of the angle of the Walker environment

Figure 7: Histograms of the quantities of interest for the Mujoco environments

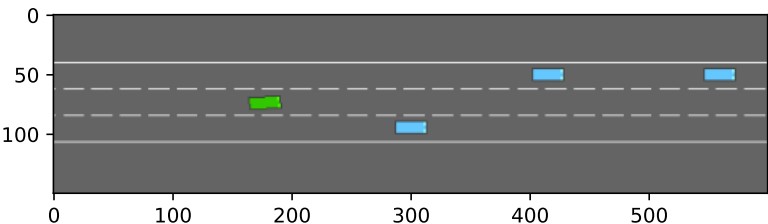

Figure 8: Simple illustration of the highway environment

We compare the discrete version of EVAC with QR-DQN (Dabney et al., 2017) on metrics such as the percentage failure, cumulative reward and the CVaR value similar to Section 8.

| Environment | Algorithm | Percentage Failure | Cum. Reward | CVaR |
|---|---|---|---|---|
| Highway | QR-DQN | $32.1 \pm 42.58$ | $133.2 \pm 80.39$ | $31.54 \pm 19.87$ |
| | EVAC | $\mathbf{0.0 \pm 0.0}$ | $\mathbf{185.27 \pm 6.59}$ | $\mathbf{37.95 \pm 16.84}$ |

Table 7: Table showing the percentage failure, cumulative reward and CVaR for the discrete action space Highway Environment

.

# K    Evolution of the percentage failure

We plot the percentage failure as a function of the number of training iterations (same as the number of data samples collected) in Figure 9. We observe that the EVAC agents exhibit lower percentage failures throughout training and understand high extreme risk regions accurately.

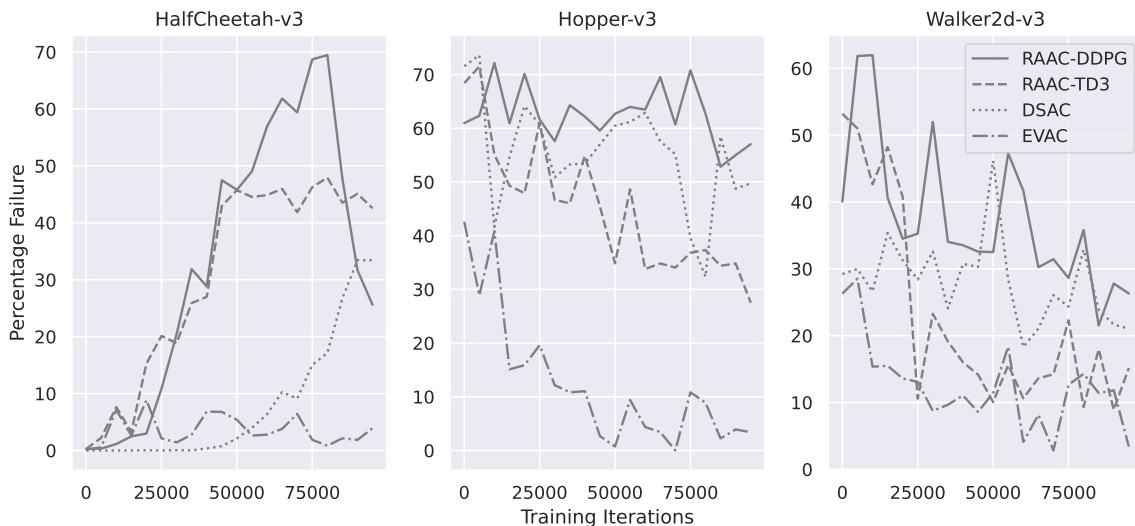

Figure 9: Figure indicating the evolution of the percentage failure as a function of number of samples/ training iterations. The EVAC agent reaches smaller percentage failure rates faster in comparison to the other baselines.

# L    Experiments with other complex risk scenarios

## L.1    Risk aversion against rare malicious attackers in a mobile communication environment

Often times when cell phone users move about in a given area, they need to connect to different transmitting base stations near them, so that they can communicate over the channel and observe high Quality of experience (QoE). Given multiple base stations and cell phone users the control problem is to assign users to different base stations so that the over all QoE is maximized. However, in such a real world setting there may exist security compromised base stations that perform stealth attacks. Stealth attacks (Jakobsson et al., 2004) are characterized by base stations that perform rare attacks on agents in a manner that remains undetectable. Such rare security compromises could lead to devastating consequences to cell phone users and need to be flagged as unsafe.

We employ mobile-env (Schneider et al., 2022), an open source environment that simulates the connections and QoE between several base stations and cell phone users. To simulate stealth attacks, we nominate one of the base stations to be malicious meaning that it very rarely causes data privacy issues for the cell phone users. The design of a risk averse policy in this scenario involves actively disconnecting from the unknown malicious base station for any cell phone user.

We simulate these stealth attacks in two scenarios for the wireless communication.

**Scenario A:** We design 4 base stations and just 3 cell phone users who randomly move about with a certain speed. There is a single malicious base station that is unknown to the agent. The observation space of the agent is 27 dimensional and consists information on the connection between the cell phone and the base station, SNR of the channel etc. The reward is rarely penalized at rate $\mathbf{p} = 5\%$ by a negative penalty for any connections made to the malicious base station. The action space is continuous valued and its dimensionality corresponds to the number of cell phones.

**Scenario B:** The setup is similar to Scenario-A except we have 4 base stations and 5 cell phone users that randomly move at a certain speed. The observation space is 45 dimensional. This set up is more challenging than the previous ones as it involves multiple cell phone users trying to connect to a limited number of base stations.

We evaluate the percentage failure (which indicates the number of times the malicious base stations was connected to in th episode), the cumulative reward and the mean CVaR below.

| Scenario | Algorithm | Percentage Failure | Cum. Reward | CVaR |
|---|---|---|---|---|
| Scenario-A | RAAC-TD3 | 38.89 ± 11.34 | -4.4 ± 2.17 | 33.46 ± 45.92 |
| | DSAC | 88.89 ± 9.07 | -1.21 ± 0.95 | 11.9 ± 14.71 |
| | EVAC | 3.7 ± 5.24 | -6.5 ± 2.3 | 34.8 ± 4.13 |
| Scenario-B | RAAC-TD3 | 59.26 ± 41.9 | -3.97 ± 1.51 | -65.75 ± 14.93 |
| | DSAC | 67.38 ± 12.94 | -2.79 ± 0.8 | 1.71 ± 14.7 |
| | EVAC | 29.63 ± 22.83 | -4.99 ± 0.97 | 37.52 ± 4.97 |

Table 8: Table showing various performance metrics for risk avoidance when malicious base stations are introduced into the mobile communication environment

.

A good risk averse policy in this setup will not connect to the malicious base station. However, being not connected to the base station reduces overall reward (QoE). Thus, we see that EVAC which mostly understands not to connect to the malicious base station (demonstrated through low percentage failures), incurs smaller cumulative rewards.

Thus, EVAC which uses EVT based tail modelling is risk averse even in the presence of rare malicious stealth attacks by a base station.

### L.2   High dimensional risk region for the Walker2d environment

The Walker environment as described in (Brockman et al., 2016) is a locomotive robot model that is tasked with walking upright with as high a speed as possible. However certain gaits of the walker may lead to potentially catastrophic consequences under certain environment settings. In this regard, we penalize the walker when it moves with very high velocity ,when its head angle is bent and when it hops instead of walking.

We monitor the velocity $v$, head angle $\theta$ and the vertical height $h$ of the walker robot from its state space. We define a hyperplane risk surface to introduce rare penalties. We define the risk hyperplane as below:

$$w_v \cdot \left(v - v_0\right) + w_\theta \cdot \left(|\theta| - \theta\right) + w_h \cdot \left(h - h_0\right) > 0$$

where $w_v, w_\theta, w_h$ indicate the importance weight for velocity, head angle and height of the walker. We set this to be a fixed $w_v = 0.3, w_\theta = 1.0, w_h = 0.5$. Any state space variable satisfying the inequality is considered

risky. Unlike, the single paramter threshold, this is more restrictive and requires the agent to keep track of multiple state space variables. We introduce rare risky penalty with Bernoulli rate $\mathbf{p} = 5\%$ and assess the risk aversion of EVAC and other baseline algorithms.

| Algorithm | Percentage Failure | Cum. Reward | CVaR |
|---|---|---|---|
| RAAC-DDPG | $13.76 \pm 5.08$ | $265.57 \pm 243.66$ | $158.18 \pm 15.82$ |
| RAAC-TD3 | $10.8 \pm 2.13$ | $201.98 \pm 106.36$ | $67.47 \pm 7.21$ |
| DSAC | $16.76 \pm 4.78$ | $566.47 \pm 127.98$ | $167.97 \pm 28.88$ |
| EVAC | $1.76 \pm 1.35$ | $267.74 \pm 51.04$ | $160.68 \pm 6.34$ |

Table 9: Table showing various performance metrics for the complex risk hypersurface
.

EVAC shows small amounts of cross over to the risky hypersurface while having lower variance in CVaR values.

## M   Limitations and Future Work

The use of EVT for risk mitigation in distributional reinforcement is novel. Our work tries to illustrate the benefit of using EVT theory in risk aware reinforcement learning. One of the challenges in the use of EVT RL is the overhead involved in training parameters of the GPD distribution through maximum likelihood estimation. The added overhead may prove expensive for both training and inference, especially on edge devices. We quantitatively discuss this computational overhead in the concluding part of Section B.We plan to further investigate the usefulness of EVT RL in other domains and study its scaling in other complex environments. Furthermore, we would like to investigate the applicability of EVAC in the offline reinforcement learning setting.

