# OpenReview forum: "Extreme Risk Mitigation in Reinforcement Learning using Extreme Value Theory"
_TMLR — Accepted by TMLR_

### Review · Reviewer_czsJ · 2024-02-27

**Summary Of Contributions:**

This paper contributes a novel approach for risk mitigation in reinforcement learning using extreme value theory. They show that prior approaches do not well model tail risk and learn a parameterized distribution of the state-action value distribution. Using this parameterization the paper proposes an actor-critic framework that can effectively optimize a policy to avoid rare extreme events. The evaluations show good empirical success and perform better than baselines. The authors study their approach in Mujoco tasks where there are rare penalties on velocities and angles and in safety gym where there are hazards that provide a rare penalty. The results show quantitatively and qualitatively that the proposed EVAC algorithm achieves the desired policy while avoiding the rare catastrophic events.

**Audience:**

Yes

**Claims And Evidence:**

Yes

**Requested Changes:**

When citing papers parenthetically \citep{} should be used rather than \cite{}.

Why just split the distribution into tail and non-tail. How do you decide where to split and what are the downsides of modeling the distribution in this way? Would it be better to model the distribution with more than 2 pieces?

The authors show that prior work on distributional RL is not sufficient for mitigating rare bad events, but then it seems that the proposed approach still relies on distributional RL. I would like clarification on how the EVT approach compares to prior work.

I would also like more details on how the different hyperparameters were chosen for the domains.

**Strengths And Weaknesses:**

Strengths:

+ This paper proposes a mathematically rigorous method for avoiding bad but rare events in RL.
+ The empirical results support the claims and the experiments are well designed.
+ The paper studies an important topic in RL

Weaknesses:
- The paper is very dense and hard to follow for a reader unfamiliar with EVT.
- It is unclear how sensitive the algorithm is to hyperparameter choices.

---

### Review · Reviewer_wbX5 · 2024-02-28

**Summary Of Contributions:**

The work presented in this paper features an approach to conservative, risk-sensitive Reinforcement Learning by modeling the tail of a state-action return distribution’s tail using extreme value theory (EVT). The proposed approach is developed to reduce the variance when estimating extremely high risk events that rarely occur, a demonstrated limitation of standard quantile regression.

This motivates a new actor-critic based approach that alternates between training the return distribution using quantile regression and then maximizing the likelihood of the extreme value distribution modeled to cover the high-risk tail behavior (using a generalized Pareto distribution). A control policy is optimized using a CVaR risk measure, seeking to minimize the selection of actions that lead to possible extreme outcomes. Altogether, this algorithmic approach (named EVAC) is tested against a collection of rudimentary benchmarks from Mujoco and Safety-Gym, adapted to test the risk-sensitivity (and risk aversion) of the proposed approach against a small collection of relevant baselines derived from two prior methods from the literatures.

In total the paper makes the following claims (to the best of my understanding):
 1. Better modeling of high-risk extreme events is provided by using EVT in place of traditional quantile regression.

 2. Variance reduction in the estimation of rare events via EVT is maintained even when data is limited.

 3. The proposed EVAC algorithm and policy optimization approach provides improved cumulative performance while significantly reducing the effect of possible extreme events through learned avoidance of a priori defined risky regions.

**Audience:**

Yes

**Claims And Evidence:**

No

**Requested Changes:**

I would highly recommend that the authors take a more careful pass through the writing and revise their statements to be more direct, clear and simple while introducing the paper, the concepts used to develop the proposed EVAC as well as when highlighting the limitations of prior work.

There needs to be proper grounding of the work and the demonstrated empirical justification in actual problems of import. Why is reducing variance of tail behavior in a distribution important? How do the experiments demonstrate this?

There should be an effort to incorporate CODAC from Ma, et al (2021) into the experimental analysis. Granted the algorithm is designed for offline RL, but so was RAAC (Urpi, et al (2021)) which didn’t restrict the authors from using it in this work as a comparative baseline.

A more thorough literature review of risk-sensitive RL in safety-critical domains. I’ve provided a set of relevant references from recent years that address relevant issues and are highly related to the approaches taken in this work.

**Strengths And Weaknesses:**

## Strengths

This paper is well organized and lays out a clear narrative that motivates the proposed approach for risk-sensitive RL. I appreciated the principled analysis used to argue for the use of extreme value theory to model the long tail of the state-action return distribution. The use of robust risk estimators such as CVaR has been well supported in the risk-sensitive RL community, this paper contributes yet another demonstration of the efficacy of using this form of objective when formulating decisions in environments that incorporate outsized negative risks.

The intended contributions and proposed claims are well framed within the context of related literature. The experiments are appropriately designed to test and analyze these claims and appropriate baselines are chosen to compare with.

Despite an extensive set of weaknesses, to be outlined and summarized next, I do believe this to be an interesting paper and of worth to the TMLR community. I am confident that it is well within the author’s capabilities to improve this work and shore up their intentions and demonstrated contributions + claims that the current paper fails to completely justify or validate.

## Weaknesses

As I go through the following list of concerns and questions I had while reviewing the paper, I have attempted to put the most serious items at the beginning of this section. Many of these can be attributed to an incomplete framing of the problem and inconsistent or partial explanation of concepts, notation or definitions. Thankfully many of these issues can be addressed through a more careful revision of the text, with particular focus on justifying and clarifying each equation and modeling assumption. Another major consideration is a lack of adequate grounding of the narrative justifying the problem setting. As currently written, this connection is suspicious and lazy. I’ll discuss more about this below. **Finally, the last major (and perhaps the most serious) issue I found with this paper is that it omits perhaps the most appropriate baseline comparison to the proposed EVAC among the literature it cites, specifically CODAC-C from Ma, et al (2021).** In addition to this, the experimental benchmarks it uses are somewhat rudimentary from a performance standpoint (even with consideration of the extremely rare, high-risk adjustments to the domains).

**I found the paper somewhat hard to follow in places as the exposition was incomplete and makes somewhat vague claims and justifications.** Perhaps it was the use of unfamiliar terminology that made it hard for me to follow. This unfortunately starts early in the paper. After introducing challenges of estimating the risk of seldom extreme events (beginning of page 2) the next paragraph refers to a “phenomenon” that hasn’t been adequately characterized or explained. Perhaps what is meant is that there are characteristics of current risk estimation approaches that produce inadequate or incomplete understanding of true risks. I do not want to belabor this point and highlight each instance throughout the paper where similar vague posturing occurs, possibly in an attempt to be more formal or make the insights appear more important than they are. It is however a pretty consistent issue throughout the current version of the paper. I would highly recommend that the authors take a more careful pass through the writing and revise their statements to be more direct, clear and simple while introducing the paper, the concepts used to develop the proposed EVAC as well as when highlighting the limitations of prior work. The paper would greatly benefit from this level of fine-grained revision.

Part of the vague writing problems throughout the paper is the absence of a clear problem statement. From what I could put together, the intention and focus of the work is only as applied to continuous control problems with full observation and dense rewards, using value-based algorithms. It would help improve the clarity of the paper if the type of RL problems the proposed methods and approach are intended for.

**The focus and importance of the problem setting used in this paper is poorly motivated and comes across as a post-hoc, hurried connotation to justify the existence of the work.** I acknowledge that this is a harsh statement from a reviewer. However, there are no real descriptions of where poor decisions in a high-risk environment (perhaps from poor modeling of the tails of the return distribution, incurring high variance?) could be detrimental. There is currently no real connection to the intended problem setting and how the proposed contributions could potentially be used to address current limitations. Without this the proposed approach and the demonstrated results are forgettable. I urge the authors to reconsider the narrative framing used to justify the need for better modeling of rare, high-risk events in safety critical scenarios. There are several clear examples of how to do this well in recent risk-sensitive RL literature (Greenberg, et al 2022; Hau, et al 2022; Liu, et al 2022; Killian, et al. 2023; Kastner, et al 2023; Leo, et al 2023) that was unfortunately omitted from this paper. Beyond being good examples, these papers are relevant to this work and possibly should be included as citations (full references at the bottom of this section).

To connect with the next paragraph, **it is concerning that the authors intentionally omit the use of CODAC-C from Ma, et al (2021) as a baseline** despite using a very similar experimental setup descried in Section 8.3. This algorithm is already demonstrated to improve beyond RAAC and D-SAC which are the only baselines used in this work. I cannot fully get behind advocating for this work without an honest effort to include CODAC-C as a baseline. Without full and adequate comparison to relevant baselines, the utility and overall performance of the proposed EVAC algorithm is called into question.

As may have been easy to tell from the previous paragraphs, I was largely underwhelmed by the experimental results. They are indeed promising and do appear to provide adequate conservatism without reducing overall performance. However, the experimental domains are purposefully crafted to reward cut and dry avoidance with pre-defined risk regions and are relatively easy control problems given the dense reward settings. The initial experimental analyses (namely the results in Figure 1a) are unsurprising, particularly given dense reward settings. The training and evaluation setup is not apparent and it is unclear how reliable the presented results in Section 8 are. **How are the training and evaluation settings different?** Are the values presented in Tables 1 and 2 drawn from the latter stages of training or from a held out set of evaluation tasks within the same domain?

I’m somewhat confused by the modeling choices made in the use of EVT to represent the long tail of the estimated return distribution. It is unclear what is meant by using values that are “large enough” for $u$, $\eta$, etc… With the emphasis on a large value for $\eta$, isn’t the effect of the GPD modeling of the extreme values somewhat minimized or ignored? It’s not clear how much the intended contributions are actually used? There is a lot left to be desired in how the method and the modeling choices are presented in Section 6. There’s a lot of emphasis on the reduction in high variance estimation of extreme events using EVT and that the use of quantile regression, focused solely on the pinball loss, is insufficient in Section 5 but this very same loss is used for EVAC?! Does this make sense? Are the anticipated weaknesses the pinball loss not fully addressed by EVAC because of the mixture distribution used to model the tail and non-tail portions of the state-action return distribution? __This calls into question the claim made at the end of page 10 in Section 7.2.1__. In total, Section 7 seems to make a lot of postulating without providing sufficient contextualization or justification/proof. This could be amended by more directly and clearly expressing the intended insights in the discussion.

In a related note, it’s somewhat confusing that the authors choose to evaluate and discount previous distributional risk-sensitive RL approaches via the Pinball loss which is not used in practice. As acknowledged by the authors, these approaches use a different loss function. If there are similar variance issues, it would be beneficial for this paper for the foundational analysis that motivates the use of EVT to be based on what is actually used in “competing” risk-sensitive RL approaches. This leads me to question whether the implementations of the baselines compared to EVAC use the pinball loss instead of their actual loss functions?

**The claim that EVAC performs well, even in the presence of limited data is not justified or demonstrated in any of the experiments.** If this is what is intended by the sparse rates of the extreme values occurring (presented in Figure 1), then this claim needs to be rephrased as it is currently misleading. There is a difference between making determinations of risk from limited data (handled in Killian, et al 2023) and relative sparsity of extreme events. I would encourage the authors to be very clear about what the intended claims are.

The first two paragraphs of Section 8 are repetitive and could be combined.

What determined the choices of the rates of penalization of the Mujoco environments? This is unclear and seems to be inconsistent between environments. Is there a principled reason behind the choices? I have a hard time relying on the relevance of the domains and the communicated results since the domains seem specifically crafted for the intended algorithmic solution… Additionally, a replication of Figure 2 in Urpi, et al (2021) would be particularly notable to demonstrate the efficacy of EVAC in avoiding the risk regions in these domains where they are clearly defined a priori and broken down by the applied continuous actions.

### References

`[Ma, et al (2022)]` Ma, Y, D Jayaraman, and OBastani. "Conservative offline distributional reinforcement learning." NeurIPS 2021.

`[Greenberg, et al (2022)]` Greenberg, I., Chow, Y., Ghavamzadeh, M., & Mannor, S. “Efficient risk-averse reinforcement learning”. NeurIPS 2022

`[Hau, et al (2022)]` Hau, J. L., Petrik, M., Ghavamzadeh, M., & Russel, R. “RASR: Risk-averse soft-robust mdps with EVAR and entropic risk”. arXiv preprint arXiv:2209.04067.

`[Liu, et al (2022)]` Liu, G., Luo, Y., Schulte, O., & Poupart, P. “Uncertainty-Aware Reinforcement Learning for Risk-Sensitive Player Evaluation in Sports Game”. NeurIPS 2022

`[Killian, et al. (2023)]` Killian, T. W., Parbhoo, S., & Ghassemi, M. “Risk Sensitive Dead-end Identification in Safety-Critical Offline Reinforcement Learning.” TMLR 2023

`[Kastner, et al (2023)]` Kastner, T., Erdogdu, M. A., & Farahmand, A. M. “Distributional Model Equivalence for Risk-Sensitive Reinforcement Learning”. NeurIPS 2023

`[Luo, et al (2023)]` Luo, Y., Liu, G., Poupart, P., & Pan, Y. “An Alternative to Variance: Gini Deviation for Risk-averse Policy Gradient”. NeurIPS 2023

---

### Review · Reviewer_CjPa · 2024-03-04

**Summary Of Contributions:**

This paper studies risk-aversion RL with a focus on the setting where low-probability but high-penalty risks exist, which poses great challenges to existing risk-aversion RL methods.

To my knowledge, the contributions of this papers are:

- This paper reveals that existing distribution-based risk-aversion RL methods struggle in modelling heavy-tail value distribution and thus lead to poor aversion of rare risks.
- This paper presents a new method to model state-action value distribution, inspired by Extreme Value Theory, through leveraging a decomposed approximation with the non-tail distribution and the tail distribution separately. This brings reduced variance and better approximation performance in accuracy and efficiency.
- The proposed method is integrated into typical AC algorithms, whose effectiveness are demonstrated in MuJoCo and Safety-gym Environments.

**Audience:**

Yes

**Claims And Evidence:**

Yes

**Requested Changes:**

1. At the top of page 5, ‘distinct thresholds $u$ for each $(s,a)$’ is mentioned. However, I do not find more information about it. The details of how it is realized/learned should be clarified.
2. The sample efficiency of the proposed method seems lacking. I find no learning curves for specific performance metric v.s. sample number.
3. Although I agree that the overhead introduced for the GPD training and sampling is not a big point. I recommend the authors to give quantitive discussion because it can be useful to readers for practical consideration.
4. I recommend the authors to discuss the potential of extending the proposed method to discrete-action and sparse-reward environments.
5. The information on how many random seeds/independent trials are used for the results are missing.

**Strengths And Weaknesses:**

###Strengthes:

- This paper is overall well-written and organized. The
- The proposed method is well motivated and supported by both theoretical analysis and empirical evaluation.
- The experiments are conducted from different perspectives, including performance comparison, risk-aversion evaluation, ablation and hyperparameter analysis.

&nbsp;

###Weaknesses:

- Some details of the proposed method are not clear.
- Sample efficiency is not presented, since the GPD distribution approximation is expected to require smaller number of samples as claimed.
- The proposed method is only evaluated in continuous control environments with dense rewards.

---

### Decision · Action_Editor_R9c6 · 2024-04-13

**Recommendation:** Accept as is

**Comment:**

This submission presents a novel and interesting use of extreme value theory (EVT) for risk-averse RL. The authors provide both theoretical and empirical justification for their work, and the presentation is clear and well-motivated. The authors have successfully addressed the reviewers concerns, so we are all in agreement to accept this work.

**Audience:**

This work would be mostly of interest to people working in risk-sensitive RL, but could also be interesting to the general RL community, as there may be value/use in EVT beyond risk-aversion.

**Claims And Evidence:**

The paper provides sufficient and accurate evidence (both theoretical and empirical) to justify their claims. After some reviewers expressed some dissatisfaction with the "hand-crafted" environments, the authors added some experiments on a mobile communication task (appendix L.1), which strengthens the empirical evidence.